# GENERALIZED TEMPORAL DIFFERENCE LEARNING MODELS FOR SUPERVISED LEARNING

## ABSTRACT

In conventional statistical learning settings, data points are typically assumed to be independently and identically distributed (i.i.d.) according to some unknown probability distribution. Various supervised learning algorithms, such as generalized linear models, are derived by making different assumptions about the conditional distribution of the response variable given the independent variables. In this paper, we propose an alternative formulation in which data points in a typical supervised learning dataset are treated as interconnected, and we model the data sampling process by a Markov reward process. Accordingly, we view the original supervised learning problem as a classic on-policy policy evaluation problem in reinforcement learning, and introduce a generalized temporal difference (TD) learning algorithm to address it. Theoretically, we establish the convergence of our generalized TD algorithms under linear function approximation. We then explore the relationship between TD's solution and the original linear regression solution. This connection suggests that the probability transition matrix does not significantly impact optimal solutions in practice and hence can be easy to design. In our empirical evaluations, we examine critical designs of our generalized TD algorithm, and demonstrate the competitive generalization performance across a variety of benchmark datasets, including regression, binary classification, and image classification within a deep learning context.

## 1 INTRODUCTION

Conventional statistical supervised learning (SL) typically assumes that the dataset is drawn from an unknown probability distribution. The primary objective in such scenarios is to learn the relationship between the features and the output (response) variable. To achieve this, generalized linear models (Nelder & Wedderburn, 1972; McCullagh & Nelder, 1989) are considered a generic algorithmic framework employed to derive objective functions. These models make specific assumptions regarding the conditional distribution of the response variable given input features, which can take forms such as Gaussian (resulting in ordinary least squares), Poisson (resulting in Poisson regression), or multinomial (resulting in logistic regression or multiclass softmax cross-entropy loss).

In recent years, reinforcement learning (RL), widely utilized in interactive learning settings, has witnessed a surge in popularity. This surge has attracted growing synergy between RL and SL, where each approach complements the other in various ways. In SL-assisted RL, the area of imitation learning (Hussein et al., 2017) may leverage expert data to regularize/speed up RL, while weakly supervised methods (Lee et al., 2020) have been adopted to constrain RL task spaces, and relabeling techniques contributed to goal-oriented policy learning (Ghosh et al., 2021).

Conversely, RL has also expanded its application into traditional SL domains. RL has proven effective in fine-tuning large language models (MacGlashan et al., 2017), aligning them with user preferences. Additionally, RL algorithms (Gupta et al., 2021) have been tailored for training neural networks (NNs), treating individual network nodes as RL agents. In the realm of imbalanced classification, RL-based control algorithms have been developed, where predictions correspond to actions, rewards are based on heuristic correctness criteria, and episodes conclude upon incorrect predictions within minority classes (Lin et al., 2020).

Existing approaches that propose a RL framework for solving supervised learning problems often exhibit a heuristic nature. These approaches involve crafting specific formulations, including ele-

ments like agents, reward functions, action spaces, and termination conditions, based on intuitive reasoning tailored to particular tasks. Consequently, they lack generality, and their heuristic nature leaves theoretical assumptions, connections between optimal RL and SL solutions, and convergence properties unclear. To the best of our knowledge, it remains uncertain whether a unified and systematic RL formulation capable of modeling a wide range of conventional supervised learning problems exists. Such a formulation should be agnostic to learning settings, including various tasks such as ordinary least squares regression, Poisson regression, binary or multi-class classification, etc. The potential benefit of using RL for predictions was first noted by Sutton (1988). They analyzed the circumstances under which a problem is better suited for solution by temporal difference (TD) learning compared to SL when both are applicable.

In this study, we introduce a generic Markov process formulation for data generation, offering an alternative to the conventional i.i.d. data assumption in SL. We also demonstrate the potential benefits of using TD algorithms. Specifically, when faced with a supervised learning dataset, we view the data points as originating from a Markov reward process (MRP) (Szepesvari, 2010). To accommodate a wide range of problems, such as Poisson regression, binary or multi-class classification, we introduce a generalized TD learning model in Section 3, whose convergence is given in Section 4 under linear function approximation. Additionally, we explore the relationship between the solutions obtained through TD learning and the original linear regression, and establish an equivalence between them under specific conditions, which renders the ease of designing the transition probability matrix. Our paper concludes with an empirical evaluation of our TD algorithm in Section 5, assessing its critical design choices and practical utility when integrated with a deep neural network across various tasks, achieving competitive results and, in some cases, improvements in generalization performance. We view our work as a step towards unifying diverse learning tasks from two pivotal domains within a single, coherent theoretical framework.

## 2 BACKGROUND

This section provides a brief overview of the fundamental concepts in statistical supervised learning and reinforcement learning settings, laying the groundwork for the introduction of our formulation and algorithm in the next section.

### 2.1 CONVENTIONAL SUPERVISED LEARNING

In the context of statistical learning, we make the assumption that data points, in the form of $(\mathbf{x}, y) \in \mathcal{X} \times \mathcal{Y}$, are independently and identically distributed (i.i.d.) according to some unknown probability distribution $P$. The goal is to find the relationship between the feature/input $\mathbf{x}$ and response/output/label variable $y$ given a training dataset $\mathcal{D} = \{(\mathbf{x}_i, y_i)\}_{i=1}^{n}$.

In a simple linear function approximation case, a commonly seen algorithm is ordinary least squares (OLS) that optimizes squared error objective function

$$\min_{\mathbf{w}} \ ||\mathbf{X}\mathbf{w} - \mathbf{y}||_2^2. \tag{1}$$

where $\mathbf{X}$ is the $n \times d$ feature matrix and $\mathbf{y}$ is the corresponding $n$-dimensional training label vector, and the $\mathbf{w}$ is the parameter vector we aim to optimize.

From a probabilistic perspective, this objective function can be derived by assuming $p(y|\mathbf{x})$ follows a Gaussian distribution with mean $\mathbf{x}^\top \mathbf{w}$ and conducting maximum likelihood estimation (MLE) for $\mathbf{w}$ with the training dataset. It is well known that $\mathbb{E}[Y|\mathbf{x}]$ is the optimal predictor (Bishop, 2006). For many other choices of distribution $p(y|\mathbf{x})$, generalized linear models (GLMs) (Nelder & Wedderburn, 1972) are commonly employed for estimating $\mathbb{E}[Y|\mathbf{x}]$. This includes OLS, Poisson regression and logistic regression, etc.

An important concept in GLMs is the *inverse link function*, which we denoted as $f$, that establishes a connection between the linear prediction (also called the **logit**), and the conditional expectation: $\mathbb{E}[Y|\mathbf{x}] = f(\mathbf{x}^\top \mathbf{w})$. For example, in logistic regression, the inverse link function is the sigmoid function. We later propose generalized TD learning models within the framework of RL that correspond to GLMs, enabling us to handle a wide range of data.

| SL Definitions | RL Definitions |
| --- | --- |
| Feature matrix $\mathbf{X}$ | Feature matrix $\mathbf{X}$ |
| Training feature of the $i$th example $\mathbf{x}_i$ | The $i$th state feature $\phi(s_i) = \mathbf{x}_i$ |
| Training target of the $i$th example $y_i$ | The state value $v(s_i) = y_i$ |

Table 1: How definitions in SL correspond to those in RL

## 2.2 REINFORCEMENT LEARNING

Reinforcement learning is often formulated within the Markov decision process (MDP) framework. An MDP can be represented as a tuple $(\mathcal{S}, \mathcal{A}, r, P, \gamma)$ (Puterman, 2014), where $\mathcal{S}$ is the state space, $\mathcal{A}$ is the action space, $r : \mathcal{S} \times \mathcal{A} \mapsto \mathbb{R}$ is the reward function, $P(\cdot|s, a)$ defines the transition probability, and $\gamma \in (0, 1]$ is the discount factor. Given a policy $\pi : \mathcal{S} \times \mathcal{A} \to [0, 1]$, the return at time step $t$ is $G_t = \sum_{i=0}^{\infty} \gamma^i r(S_{t+i}, A_{t+i})$, and value of a state $s \in \mathcal{S}$ is the expected return starting from that state $v^\pi(s) = \mathbb{E}_\pi[G_t | S_t = s]$. In this work, we focus on the policy evaluation problem for a fixed policy, thus the MDP can be reduced to a Markov reward process (MRP) (Szepesvari, 2010) described by $(\mathcal{S}, r^\pi, P^\pi, \gamma)$ where $r^\pi(s) \overset{\text{def}}{=} \sum_a \pi(a|s)r(s, a)$ and $P^\pi(s'|s) \overset{\text{def}}{=} \sum_a \pi(a|s)P(s'|s, a)$. When it is clear from the context, we will slightly abuse notations and ignore the superscript $\pi$.

In policy evaluation problem, the objective is to estimate the state value function of a fixed policy $\pi$ by using the trajectory $s_0, r_1, s_1, r_2, s_2, ...$ generated from $\pi$. Under linear function approximation, the value function is approximated by a parametrized function $v(s) \approx \phi(s)^\top \mathbf{w}$ with parameters $\mathbf{w}$ and some fixed feature mapping $\phi : \mathcal{S} \mapsto \mathbb{R}^d$ where $d$ is the feature dimension. Note that the state value satisfies the Bellman equation

$$v(s) = r(s) + \gamma \mathbb{E}_{S' \sim P(\cdot|s)}[v(S')]. \tag{2}$$

One fundamental approach for the evaluation problem is the temporal difference (TD) learning, which uses a sampled transition $s_t, r_{t+1}, s_{t+1}$ to update the parameters $\mathbf{w}$ through stochastic fixed point iteration based on (2) with a step-size $\alpha > 0$:

$$\mathbf{w} \leftarrow \mathbf{w} + \alpha(y_{t,td} - \phi(s_t)^\top \mathbf{w})\phi(s_t) \quad \text{where} \quad y_{t,td} \overset{\text{def}}{=} r_{t+1} + \gamma\phi(s_{t+1})^\top \mathbf{w}. \tag{3}$$

In linear function approximation setting, TD converges to the solution that solves the system $\mathbf{Aw} = \mathbf{b}$ (Bradtke & Barto, 1996; Tsitsiklis & Van Roy, 1997), where

$$\mathbf{A} = \mathbb{E}[\phi(s_t)(\phi(s_t) - \gamma\phi(s_t))^\top] = \mathbf{X}^\top \mathbf{D}(\mathbf{I} - \gamma\mathbf{P})\mathbf{X} \quad \text{and} \quad \mathbf{b} = \mathbb{E}[r_t\phi(s_t)] = \mathbf{X}^\top \mathbf{Dr} \tag{4}$$

with $\mathbf{X} \in \mathbb{R}^{|\mathcal{S}| \times d}$ being the feature matrix whose rows are the state features $\phi(s_t)$, $\mathbf{D} \in \mathbb{R}^{|\mathcal{S}| \times |\mathcal{S}|}$ being the diagonal matrix with the stationary distribution probabilities on the diagonal, $\mathbf{P} \in \mathbb{R}^{|\mathcal{S}| \times |\mathcal{S}|}$ being the transition probability matrix (i.e., $\mathbf{P}_{ij} = P(s_j|s_i)$) and $\mathbf{r} \in \mathbb{R}^{|\mathcal{S}|}$ being the reward vector.

## 3 GENERALIZED TEMPORAL DIFFERENCE LEARNING

This section describes our MRP construction given a supervised learning dataset $\mathcal{D} = \{(\mathbf{x}_i, y_i)\}_{i=1}^n$ and proposes our generalized TD learning algorithm to solve it.

**Regression**. We start by considering the basic regression setting with linear models before introducing our generalized TD algorithm. Table 1 summarizes how we can view concepts in the conventional supervised learning from an RL perspective. The key is to treat the original training label as a state value that we are trying to learn, and then the reward function can be derived from the Bellman equation (2) as

$$r(s) = v(s) - \gamma \mathbb{E}_{S' \sim P(\cdot|s)}[v(S')]. \tag{5}$$

We will discuss the choice of $P$ later. At each iteration (or time step in RL), the reward can be approximated using a stochastic example. For instance, assume that at iteration $t$ (i.e., time step in RL), we obtain an example $(\mathbf{x}_i^{(t)}, y_i^{(t)})$. We use superscripts and subscripts to denote that the $i$th training example is sampled at the $t$th time step. Then the next example $(\mathbf{x}_j^{(t+1)}, y_j^{(t+1)})$ is sampled according to $P(\cdot|\mathbf{x}_i^{(t)})$ and the reward can be estimated as $r^{(t+1)} = y_i^{(t)} - \gamma y_j^{(t+1)}$ by approximating the expectation in Eq. (5) with a stochastic example. As one might notice that, in a sequential setting the $t$ is monotonically increasing, hence we will simply use a simplified notation $(\mathbf{x}_t, y_t)$ to denote the training example sampled at time step $t$.

We now summarize and compare the updating rules in conventional SL and in our TD algorithm under linear function approximation. At time step $t$, the **conventional updating rule** based on stochastic gradient descent (SGD) is given by

$$\mathbf{w} \leftarrow \mathbf{w} + \alpha(y_t - \mathbf{x}_t^\top \mathbf{w})\mathbf{x}_t, \tag{6}$$

while our **TD updating rule** is

$$\mathbf{w} \leftarrow \mathbf{w} + \alpha(y_{t,td} - \mathbf{x}_t^\top \mathbf{w})\mathbf{x}_t \tag{7}$$

$$\text{where } y_{t,td} \overset{\text{def}}{=} r_{t+1} + \gamma\widehat{y_{t+1}} = y_t - \gamma y_{t+1} + \gamma\mathbf{x}_{t+1}^\top \mathbf{w} \tag{8}$$

and $\mathbf{x}_{t+1} \sim P(\cdot|\mathbf{x}_t)$ with ground-truth label $y_{t+1}$. The critical difference is that TD uses a bootstrap, so it does not cancel the $\gamma y_{t+1}$ term from the reward when computing the TD training target $y_{t,td}$. By setting $\gamma = 0$, one recovers the original supervised learning updating rule (6).

**Generalized TD: An extension to general learning tasks**. A natural question regarding TD is how to extend it to different types of data, such as those with counting, binary, or multiclass labels. TD is not derived by making probabilistic assumptions about the value conditioned on a state. A naïve solution would be to treat all labels in the same manner as described above. However, this approach lacks justification and does not perform well empirically, as demonstrated in Section 5.

Recall that in generalized linear models (GLMs), it is assumed that the output variable $y \in \mathcal{Y}$ follows an exponential family distribution. In addition, there exists an *inverse link function $f$* that maps a linear prediction $z \overset{\text{def}}{=} \mathbf{w}^\top \mathbf{x}$ to the output/label space $\mathcal{Y}$ (i.e., $f(z) \in \mathcal{Y}, \forall z \in \mathbb{R}$). Examples of GLMs include linear regression (where $y$ follows a normal/Gaussian distribution, $f$ is the identity function and the loss is the squared loss) and logistic regression (where $y$ is Bernoulli, $f$ is the sigmoid function and the loss is the log loss). More generally, the output may be in higher dimensional space and both $z$ and $y$ will be vectors instead of scalars. As an example, multinomial regression uses the softmax function $f$ to convert a vector $\mathbf{z}$ to another vector $\mathbf{y}$ in the probability simplex. Interested readers can refer to Banerjee et al. (2005); Helmbold et al. (1999); McCullagh & Nelder (1989, Table 2.1) for more details. As per convention, we refer to the variable $z$ as **logit**.

The significance of the logit $z$ is that it is naturally *additive*, which mirrors the additive nature of returns (cumulative sum of rewards) in RL. It also implies that one can add two linear predictions and the resultant $z = z_1 + z_2$ can still be transformed to a valid output $f(z) \in \mathcal{Y}$. In contrast, adding two labels does not necessarily produce a valid label $y_1 + y_2 \notin \mathcal{Y}$. Therefore, the idea is to construct a bootstrapped target in the real line (logit space, or $z$-space)

$$z_{t,td} \overset{\text{def}}{=} r_{t+1} + \gamma\widehat{z_{t+1}} = (z_t - \gamma z_{t+1}) + \gamma\mathbf{x}_{t+1}^\top \mathbf{w}$$

and then convert back to the original label space to get the TD target $y_{t,td} = f(z_{t,td})$. In multiclass classification problems, we often use a one-hot vector to represent the original training target. For instance, in the case of MNIST, the target is a ten-dimensional one-hot vector. Consequently, the reward becomes a vector, with each component corresponding to a class. This can be interpreted as evaluating the policy under ten different reward functions in parallel and selecting the highest value for prediction.

Algorithm 1 provides the pseudo-code of our algorithm. At time step $t$, the process begins by sampling the state $\mathbf{x}_t$, and then we sample the next state according to the predefined $P$. The reward is computed as the difference in logits after converting the original labels $y_t, y_{t+1}$ into the logit space with the link function. Subsequently TD bootstrap target is constructed in the logit space. Finally, the TD target is transformed back to the original label space before it is used to calculate the loss. Note that in standard regression is a special case where the (inverse) link function is simply the identity function, so it reduces to the standard update (6) with squared loss. In practice, we might need some smoothing parameter when the function $f^{-1}$ goes to infinity. For example, in binary classification, $\mathcal{Y} = \{0, 1\}$ and the corresponding logits are $z = -\infty$ and $z = \infty$. To avoid this, we subtract/add some small value to the label before applying $f^{-1}$.

## 4 THEORETICAL JUSTIFICATIONS

In this section, we present convergence proofs for our generalized TD algorithm (Algorithm 1) under both the expected updating rule and the sample-based updating rule. We also explore the relationship between the solutions obtained by our algorithm and conventional linear regression. Detailed proofs are provided in the Appendix A.

---

**Algorithm 1** Generalized TD for SL

---

**Input:** A dataset $\mathcal{D}$; randomly sample a data point $(\mathbf{x}_t, y_t) \in \mathcal{D}$ as the starting point. (One can also use mini-batch starting points in NNs.)
**for** $t = 1, 2, \ldots$ **do**
    Sample $\mathbf{x}_{t+1}$ according to predefined $P(\cdot|\mathbf{x}_t)$, let $y_{t+1}$ be its label
    $r_{t+1} = f^{-1}(y_t) - \gamma f^{-1}(y_{t+1})$ // Convert to logits and compute reward
    $z_{t,td} = r_{t+1} + \gamma \mathbf{x}_{t+1}^{\top} \mathbf{w}$ // Bootstrap target, a separate target network is needed in deep NNs
    $\mathbf{w} \leftarrow \mathbf{w} - \alpha \nabla l(f(z_{t,td}), y_t)$ // Convert the logit back to original label space
    // When using linear models, the update is $\mathbf{w} \leftarrow \mathbf{w} - \alpha(y_t - f(z_{t,td}))\mathbf{x}_t$

---

### 4.1 CONVERGENCE

Here we show the finite-time convergence when using our TD(0) updates with linear function approximation. We primarily follow the convergence framework presented in Bhandari et al. (2018), making nontrivial adaptations due to the presence of the inverse link function. Let $z(s) = \phi(s)^{\top}\mathbf{w}$, or $z = \phi^{\top}\mathbf{w}$ for conciseness.

**Assumption 1** (Feature regularity). $\forall s \in \mathcal{S}, \|\phi(s)\|_2 \leq 1$ *and the steady-state covariance matrix* $\mathbf{\Sigma} \stackrel{\text{def}}{=} D(s)\phi(s)\phi(s)^{\top}$ *has full rank.*

It is natural to consider the existence of a stationary point before demonstrating convergence. Assumption 1 is a typical assumption necessary for the existence of the fixed point $\mathbf{w}^*$ when there is no transform function (Tsitsiklis & Van Roy, 1997).

**Assumption 2.** *The inverse link function $f$ is continuous, invertible and strictly increasing in its domain.*

Assumption 2 is satisfied for those inverse link functions commonly used in GLMs, including but not limit to identity function (linear regression), exponential function (Poisson regression), and sigmoid function (logistic regression) (McCullagh & Nelder, 1989).

**Remark.** These two assumptions ensure that $\mathbf{w}^*$ is also the fixed point in our setting. Then we can consider a bounded domain $\mathcal{W}$ such that $\mathbf{w}^* \in \mathcal{W}$, and the following lemma holds.

**Lemma 1.** *Under Assumption 1 and $\mathbf{w} \in \mathcal{W}$, there exists $L \geq 1$ such that $\forall s_1, s_2 \in \mathcal{S}$ with $z_1 = \phi(s_1)^{\top}\mathbf{w}, z_2 = \phi(s_2)^{\top}\mathbf{w}$*

$$\frac{1}{L}|z_1 - z_2| \leq |f(z_1) - f(z_2)| \leq L|z_1 - z_2|. \tag{9}$$

This is due to $f$ being invertible and both the features and the parameters are bounded. The next assumption is necessary later to ensure that the step size is positive.

**Assumption 3** (Bounded discount). *The discount factor satisfies $\gamma < \frac{1}{L^2}$ for the $L$ in Lemma 1.*

**Convergence under expected update**. The expected update rule is given by

$$\mathbf{w}_{t+1} = \mathbf{w}_t + \alpha \overline{g}(\mathbf{w}_t) \quad \text{with} \quad \overline{g}(\mathbf{w}_t) \stackrel{\text{def}}{=} \mathbb{E}[(y_{td} - y)\phi] \tag{10}$$

where $y \stackrel{\text{def}}{=} f(z), y_{td} \stackrel{\text{def}}{=} f(z_{td}) = f(r + \gamma \phi'^{\top}\mathbf{w}_t)$ and $\phi' \stackrel{\text{def}}{=} \phi(s')$ for short.

One can expand the distance from $\mathbf{w}_{t+1}$ to $\mathbf{w}^*$ as

$$\|\mathbf{w}^* - \mathbf{w}_{t+1}\|_2^2 = \|\mathbf{w}^* - \mathbf{w}_t\|_2^2 - 2\alpha(\mathbf{w}^* - \mathbf{w}_t)^{\top}\overline{g}(\mathbf{w}_t) + \alpha^2\|\overline{g}(\mathbf{w}_t)\|_2^2 \tag{11}$$

The common strategy is to make sure that the second term can outweigh the third term on the RHS so that $\mathbf{w}_{t+1}$ can get closer to $\mathbf{w}^*$ than $\mathbf{w}_t$ in each iteration. This can be achieved by choosing an appropriate step size as shown below:

**Theorem 1.** *[Convergence with Expected Update] Under Assumption 1-3, consider the sequence $(\mathbf{w}_0, \mathbf{w}_1, \cdots)$ satisfying Eq. (10). Let $\overline{\mathbf{w}}_T \stackrel{\text{def}}{=} \frac{1}{T}\sum_{t=0}^{T-1}\mathbf{w}_t, \overline{z}_T = \phi^{\top}\overline{\mathbf{w}}_T$ and $z^* = \phi^{\top}\mathbf{w}^*$. By choosing $\alpha = \frac{1-\gamma L^2}{4L^3} > 0$, we have*

$$\mathbb{E}\left[(z^* - \overline{z}_T)^2\right] \leq \left(\frac{2L^2}{1-\gamma L^2}\right)^2 \frac{\|\mathbf{w}^* - \mathbf{w}_0\|_2^2}{T} \tag{12}$$

$$\|\mathbf{w}^* - \mathbf{w}_T\|_2^2 \leq \exp\left(-T\omega\left(\frac{1-\gamma L^2}{2L^2}\right)^2\right)\|\mathbf{w}^* - \mathbf{w}_0\|_2^2 \tag{13}$$

Eq. (12) shows that in expectation, the average prediction converges to the true value in the $z$-space, while Eq. (13) shows that the last iterate converges to the fixed point exponentially fast when using expected update. In practice, sample-based update is preferred and we discuss its convergence next.

**Convergence under sample-based update**. This subsection shows the convergence under i.i.d. sample setting. Suppose $s_t$ is sampled from the stationary distribution $D(s)$ and $s_{t+1} \sim P(\cdot|s_t)$. For conciseness, let $\phi_t \stackrel{\text{def}}{=} \phi(s_t)$ and $\phi_{t+1} \stackrel{\text{def}}{=} \phi(s_{t+1})$. Then the sample-based update rule is

$$\mathbf{w}_{t+1} = \mathbf{w}_t + \alpha_t g_t(\mathbf{w}_t) \quad \text{with} \quad g_t(\mathbf{w}_t) \stackrel{\text{def}}{=} (y_{t,td} - y_t)\phi_t \tag{14}$$

where $y_t \stackrel{\text{def}}{=} f(z_t) = f(\phi_t^\top \mathbf{w}_t)$ and $y_{t,td} \stackrel{\text{def}}{=} f(z_{t,td}) = f(r_{t+1} + \gamma\phi_{t+1}^\top \mathbf{w}_t)$.

To account for the randomness, let $\sigma^2 \stackrel{\text{def}}{=} \mathbb{E}[\|g_t(\mathbf{w}^*)\|_2^2]$, the variance of the TD update at the fixed point $\mathbf{w}^*$ under the stationary distribution. The following theorem shows the convergence when using i.i.d. sample for the update:

**Theorem 2.** *[Convergence with Sampled-based Update] Under Assumption 1-3, with sample-based update Eq. (14) and $\sigma^2 = \mathbb{E}[\|g_t(\mathbf{w}^*)\|_2^2]$. Let $\overline{\mathbf{w}}_T \stackrel{\text{def}}{=} \frac{1}{T}\sum_{t=0}^{T-1} \mathbf{w}_t$, $\overline{z}_T = \phi^\top\overline{\mathbf{w}}_T$ and $z^* = \phi^\top\mathbf{w}^*$. For $T \geq \frac{64L^6}{(1-\gamma L^2)^2}$ and a constant step size $\alpha_t = 1/\sqrt{T}, \forall t$, we have*

$$\mathbb{E}\left[(z^* - \overline{z}_T)^2\right] \leq \frac{L\left(\|\mathbf{w}^* - \mathbf{w}_0\|_2^2 + 2\sigma^2\right)}{\sqrt{T}(1 - \gamma L^2)}. \tag{15}$$

This shows that the generalized TD update converges even when using sample-based update. Not surprisingly, it is slower than using the expected update (12) in terms of the number of iterations $T$.

## 4.2 MIN-NORM SOLUTION EQUIVALENCE

This subsection provides sufficient condition for the equivalence between the minimum norm solutions of TD and OLS:

$$\mathbf{w}_{TD} = \mathbf{A}^\dagger \mathbf{b} \qquad \mathbf{w}_{LS} = \mathbf{X}^\dagger \mathbf{y} \tag{16}$$

using notations from Eq. (4). In our context, the reward is $\mathbf{r} = (\mathbf{I} - \gamma\mathbf{P})\mathbf{y}$. To simplify notations, define $\mathbf{S} \stackrel{\text{def}}{=} \mathbf{D}(\mathbf{I} - \gamma\mathbf{P})$. Then

$$\mathbf{A} = \mathbf{X}^\top\mathbf{S}\mathbf{X}, \qquad \mathbf{b} = \mathbf{X}^\top\mathbf{S}\mathbf{y}, \qquad \mathbf{w}_{TD} = \mathbf{A}^\dagger\mathbf{b} = (\mathbf{X}^\top\mathbf{S}\mathbf{X})^\dagger \cdot \mathbf{X}^\top S\mathbf{y} \tag{17}$$

Note that, it is evident that the OLS solution aligns with TD's solution under our formulation: any solution to the original system is also a solution to TD. To establish exact equivalence, we need the following assumption:

**Assumption 4.** $\mathbf{D}$ *has full support and* $\mathbf{X}$ *has linearly independent rows.*

These conditions may be easily satisfied, for example, when the transition probability matrix $\mathbf{P}$ is irreducible and in the over-parametrization regime. When $\mathbf{D}$ has full support, $\mathbf{S}$ is invertible (thus has linearly independent rows/columns). Additionally, $\mathbf{X}$ has linearly independent rows so $(\mathbf{X}^\top\mathbf{S}\mathbf{X})^\dagger = \mathbf{X}^\dagger\mathbf{S}^{-1}(\mathbf{X}^\top)^\dagger$ (Greville, 1966, Thm.3) and the min-norm solution becomes

$$\mathbf{w}_{TD} = \mathbf{A}^\dagger\mathbf{b} = \mathbf{X}^\dagger\mathbf{S}^{-1}(\mathbf{X}^\top)^\dagger\mathbf{X}^\top\mathbf{S}\mathbf{y} \tag{18}$$

Finally, when $\mathbf{X}$ has linearly independent rows, $(\mathbf{X}^\top)^\dagger\mathbf{X}^\top = \mathbf{I}_n$ so $\mathbf{w}_{TD} = \mathbf{X}^\dagger\mathbf{y} = \mathbf{w}_{LS}$. It should be noted that, the minimum norm property of TD's solution under overparameterization regime is not new, and has been discussed by Xiao et al. (2022).

**Empirical verification.** The above equivalence, despite its simplicity and intuitiveness, underscores the ease of selecting a $\mathbf{P}$. Table 2 illustrates the striking similarity between the closed-form solutions of our linear TD method and standard OLS under various choices of the probability matrix by using a synthetic dataset (details are in Appendix B.1). Two key observations emerge: 1) As the feature dimension increases towards the overparameterization regime, both solutions become nearly indistinguishable, implying that designing $\mathbf{P}$ may be straightforward when employing a powerful

| Transition \ Dimension | 70 | 90 | 110 | 130 |
|---|---|---|---|---|
| Random | $0.027$ $(\pm 0.01)$ | $0.075$ $(\pm 0.02)$ | $\leq 10^{-10}$ $(\pm 0.00)$ | $\leq 10^{-10}$ $(\pm 0.00)$ |
| Uniform | $0.026$ $(\pm 0.01)$ | $0.074$ $(\pm 0.01)$ | $\leq 10^{-10}$ $(\pm 0.00)$ | $\leq 10^{-10}$ $(\pm 0.00)$ |
| Distance (Far) | $0.028$ $(\pm 0.01)$ | $0.075$ $(\pm 0.01)$ | $\leq 10^{-10}$ $(\pm 0.00)$ | $\leq 10^{-10}$ $(\pm 0.00)$ |
| Distance (Close) | $0.182$ $(\pm 0.10)$ | $0.249$ $(\pm 0.04)$ | $\leq 10^{-10}$ $(\pm 0.00)$ | $\leq 10^{-10}$ $(\pm 0.00)$ |
| Deficient | $0.035$ $(\pm 0.01)$ | $0.172$ $(\pm 0.04)$ | $0.782$ $(\pm 0.18)$ | $0.650$ $(\pm 0.25)$ |

Table 2: Distance between closed-form min-norm solutions of TD(0) and LS $\|\mathbf{w}_{TD} - \mathbf{w}_{LS}\|_2$. Input matrix $\mathbf{X}$ has normally distributed features with dimension $d \in \{70, 90, 110, 130\}$. Results are average over 10 runs with standard error in bracket. Details can be found in Appendix B.1.

model like NN. 2) Deficient choices for $\mathbf{P}$ with non-full support can pose issues and should be avoided. In practice, deep learning settings are prevalent for real-world problems, making it feasible to opt for a computationally and memory-efficient $\mathbf{P}$, such as a uniform constant matrix where every entry is set to $1/n$. Such a matrix is ergodic and, therefore, not deficient. We will delve deeper into the selection of $\mathbf{P}$ in the next section. In practice, we have observed that the solution of TD is nearly identical to that of the OLS even when the sufficient conditions do not hold. Detailed results on real-world datasets can be found in Table 7 of Appendix B.2.

## 5 EMPIRICAL STUDIES

This section focuses on the following aspects: 1) Investigating the effectiveness of critical components of our algorithm, including the bootstrap, transition probability, and the inverse link function; 2) Analyzing the sensitivity of hyperparameters in our algorithm; 3) Assessing the practical utility when applied with relatively larger NNs. Details for reproducible research and additional results can be found in Appendix B.2.

**Setup overview.** We use fully connected NN for regression and binary classification tasks and use CNNs for image tasks. For regression with real targets, we adopt three datasets: house (Lichman, 2015), CTPOS (Graf et al., 2011) and execution time (Paredes & Ballester-Ripoll, 2018). For Poisson regression, we use Bikeshare data (Fanaee-T & Gama, 2013), where the target variable represents bike rental counts. For binary classification, we employ Australian weather (Joe Young , Owner), Cancer (Wolberg et al., 1995) and travel insurance (Company, 2021). These relatively small datasets allow for thorough performance evaluation. For image classification, we use the popular MNIST (LeCun et al., 2010), Fashion-MNIST (Xiao et al., 2017), Cifar10 and Cifar100 (Krizhevsky, 2009) datasets. In TD algorithms, unless otherwise specified, we use a fixed transition probability matrix with all entries set to $1/n$. This choice simplifies memory management, generation, and sampling processes.

**Baselines and naming rules.** Unless otherwise specified, the following baselines and naming conventions apply. TDReg: Used for continuous target values, with its direct competitor being Reg (conventional least square regression). Reg-WP: Utilizes the same probability transition matrix as TDReg but does not employ bootstrap targets. This baseline can be used to assess the effect of bootstrap and transition probability matrix. TDPoisson: Designed for handling counting data, contrasting with the vanilla Poisson regression. To evaluate the necessity of using the inverse link function $f$, we introduce the baseline with the suffix -WOF (without the function $f$).

**The effect of bootstrap**. In this set of experiments, we aim to examine the effect of bootstrap. To do so, we focus on comparing TDReg and Reg-WP, as the bootstrap is the only distinction between them in the context of regression datasets. We design experiments where we introduce zero-mean Gaussian noise with increasing variance to the training targets while leaving the test data untouched. Table 3 shows across most datasets, TD demonstrates a mild improvement in test error. As the noise increases, the benefits of TD become more evident. Intuitively, the usage of bootstrap may prevent the model from overfitting to the original training targets in the datasets. To gain further insights, we visualize the learning curves in Figure 1. Figure 1(d) illustrates that TD exhibits only mild overfitting behavior compared to the baselines during the early stages of learning.

**The hyperparameter sensitivity**. In deep learning, TD algorithms can pose challenges due to the interplay of two additional hyperparameters: the discount rate $\gamma$ and the target NN moving rate $\tau$ (Mnih et al., 2015). Optimizing these hyperparameters can often be computationally expensive. Therefore, we investigate their impact on performance by varying $\gamma, \tau$. We find that selecting appro-

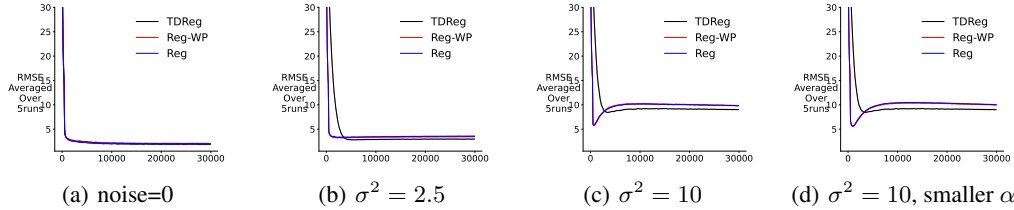

| | | | |
|:---:|:---:|:---:|:---:|
| (a) noise=0 | (b) $\sigma^2 = 2.5$ | (c) $\sigma^2 = 10$ | (d) $\sigma^2 = 10$, smaller $\alpha$ |

Figure 1: These learning curves represent the performance across different standard deviations $\sigma$ of Gaussian noise on the CTPOS dataset. The noise is only added to the training data, and the test root mean square error (RMSE) is plotted as a function of the number of mini-batch updates. The rightmost figure uses a smaller learning rate. The results have been smoothed using a 5-point window before averaging over 5 runs.

| CTPOS | | |
|:---:|:---:|:---:|
| **TD-Reg** | **Reg-WP** | **Reg** |
| $1.9230 \pm 0.1146$ | $2.0573 \pm 0.0813$ | $2.0148 \pm 0.1334$ |
| $2.9717 \pm 0.0708$ | $3.4708 \pm 0.1111$ | $3.5696 \pm 0.0714$ |
| $9.0226 \pm 0.0607$ | $9.8214 \pm 0.0851$ | $9.8328 \pm 0.0407$ |

| house | | |
|:---:|:---:|:---:|
| **TD-Reg** | **Reg-WP** | **Reg** |
| $3.3843 \pm 0.2142$ | $3.3553 \pm 0.2269$ | $3.3193 \pm 0.1630$ |
| $4.3256 \pm 0.2345$ | $4.7508 \pm 0.2452$ | $4.7551 \pm 0.2797$ |
| $10.7322 \pm 0.2835$ | $11.9900 \pm 0.5564$ | $12.1484 \pm 0.4733$ |

| Bikeshare | | |
|:---:|:---:|:---:|
| **TD-Poisson** | **Poisson-WP** | **Poisson** |
| $40.6562 \pm 0.7763$ | $40.9044 \pm 0.3671$ | $40.4971 \pm 0.4546$ |
| $40.9761 \pm 0.2627$ | $43.9063 \pm 0.4340$ | $43.9376 \pm 0.4436$ |
| $42.7952 \pm 0.2095$ | $44.8175 \pm 0.4478$ | $44.7051 \pm 0.4201$ |

Table 3: Test root mean squared error (RMSE) with standard error. Each table comprises three rows, with increasing variance of Gaussian noise ($\sigma^2 \in \{0, 2.5, 10\}$) added to the training target to observe overfitting. The results have been smoothed using a 5-point window before averaging over 5 runs.

priate parameters tends to be relatively straightforward. Figure 2 displays the testing performance across various parameter settings. It is evident that performance does not vary significantly across settings, indicating that only one hyperparameter or a very small range of hyperparameters needs to be considered during the hyperparameter sweep.

**The impact of transition probability matrix $\mathbf{P}$**. Figure 2 illustrates the sensitivity analysis when employing three intuitive types of transition matrices: $P(\mathbf{x}'|\mathbf{x})$ is larger when the two points $\mathbf{x}, \mathbf{x}'$ are 1) similar (denoted as $P_s$); 2) far apart ($P_f$); 3) $P(\mathbf{x}'|\mathbf{x}) = 1/n, \forall \mathbf{x}, \mathbf{x}' \in \mathcal{X}$ ($P_c$). The rationale for choosing these three options is as follows: the first two may lead to a reduction in the variance of the bootstrap estimate if two consecutive points are positively or negatively correlated. The last choice is consistently used due to its computational and memory efficiency and its natural simplicity. To expedite computations, $P_s, P_f$ are computed based on the training targets instead of the features. The resulting matrix may not be a valid stochastic matrix, we use DSM projection (Wang et al., 2010) to turn it into a valid one. We refer readers to Appendix B.3 for details.

Since our results in Table 2 and Figure 2 indicate that $\mathbf{P}$ does not significantly impact regular regression, we conducted experiments on binary classification tasks and observed their particular utility when dealing with imbalanced labels. We define $P_s$ by defining the probability of transitioning to the same class as $0.9$ and to the other class as $0.1$. Table 4 presents the reweighted balanced results for two binary classification datasets with class imbalance. It is worth noting that in such cases, Classify-WP serves as both 1) no bootstrap baseline and 2) the upsampling techniques for addressing class imbalance in the literature (Kubát & Matwin, 1997).

Observing that TD-Classify and Classify-WP yield nearly identical results and Classify (without using TD's sampling) is significantly worse, suggesting that the benefit of TD arises from the sampling distribution rather than the bootstrap estimate in the imbalanced case. Furthermore, $P_f, P_s$ yield almost the same results in this scenario since they provide the same stationary distribution (equal weight to each class), so here Classify-WP represents both. We also conducted tests using $P_c$, which yielded results that are almost the same as Classify, and have been omitted from the table. In conclusion, the performance difference of TD in the imbalanced case arises from the tran-

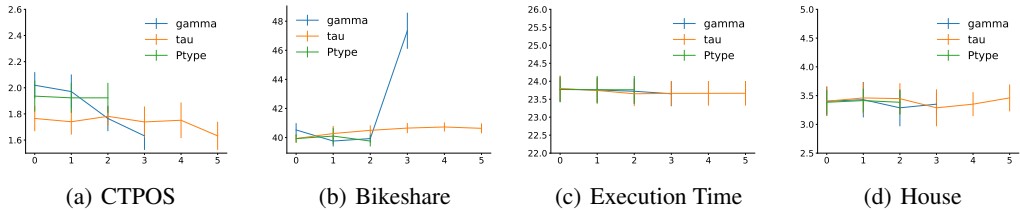

| (a) CTPOS | (b) Bikeshare | (c) Execution Time | (d) House |

Figure 2: Sensitivity to hyperparameter settings. We show test RMSE with error bars while sweeping over discount rate $\gamma \in \{0.1, 0.2, 0.4, 0.9\}$, target NN moving rate $\tau \in \{0.001, 0.01, 0.1, 0.2, 0.4, 0.9\}$, and three types of transition probability. $\gamma$ significantly hurts only when it goes to the largest value on bikeshare, and it is likely because the exponential term in Poisson regression needs a unusually small learning rate. When generating curves for $\tau$ and $\gamma$, we maintain $P_{\text{type}}$ as a simple uniform constant, further indicating that sweeping over this hyperparameter is unnecessary.

sition probability matrix rather than the bootstrap target. The transition matrix's impact is due to the implied difference in the stationary distribution.

| Algs
Dataset | TD-Classify | Classify-WP | Classify | TD-WOF |
|---|---|---|---|---|
| **Insurance** | $0.0073 \pm 0.0001$ | $0.0073 \pm 0.0001$ | $0.0144 \pm 0.0003$ | $0.4994 \pm 0.0005$ |
| **Weather** | $0.0695 \pm 0.0008$ | $0.0701 \pm 0.0008$ | $0.0913 \pm 0.0010$ | $0.4965 \pm 0.0031$ |
| **Cancer** | $0.0231 \pm 0.0023$ | $0.0241 \pm 0.0021$ | $0.0244 \pm 0.0034$ | $0.4883 \pm 0.0105$ |

Table 4: Binary classification with imbalance. 0.0073 means 0.73% misclassification rate, or $1 - 0.73\% = 99.27\%$ accuracy. The results are smoothed over 5 evaluations before averaging over 5 random seeds.

**The usage of inverse link function**. The results of TD on classification without using a transformation are presented in Table 4 and are marked by the suffix 'WOF.' These results are not surprising, as the bootstrap estimate can potentially disrupt the TD target entirely. Consider a simple example where a training example $\mathbf{x}$ has a label of one and transitions to another example, also labeled one. Then the reward ($r = y - \gamma y'$) will be $1 - \gamma$. If the bootstrap estimate is negative, the TD target might become close to zero or even negative, contradicting the original training label of one significantly.

**Image dataset**. For image datasets, we employed a convolutional neural network (CNN) architecture consisting of three convolution layers with the number of kernels $32$, $64$, and $64$, each with a filter size of $2 \times 2$. This was followed by two fully connected hidden layers with $256$ and $128$ units, respectively, before the final output layer. The error rates are presented in Table 5. Our TD algorithm demonstrated competitive performance when compared to the classical approach.

| Algs
Dataset | TD-Classify | Classify-WP | Classify |
|---|---|---|---|
| **mnist** | $0.0094 \pm 0.0002$ | $0.0103 \pm 0.0006$ | $0.0100 \pm 0.0005$ |
| **mnistfashion** | $0.1090 \pm 0.0006$ | $0.1074 \pm 0.0013$ | $0.1104 \pm 0.0001$ |
| **cifar10** | $0.3258 \pm 0.0015$ | $0.3289 \pm 0.0041$ | $0.3287 \pm 0.0006$ |
| **cifar100** | $0.6845 \pm 0.0015$ | $0.6874 \pm 0.0043$ | $0.6879 \pm 0.0037$ |

Table 5: Image classification test error. 0.0094 means 0.94% misclassifications, or $1 - 0.94\% = 99.06\%$ accuracy. The results are smoothed over 10 evaluations before averaging over 3 random seeds.

## 6 CONCLUSION

This paper presents a general formulation that converts classic SL problems into RL problems, employing a generalized TD algorithm for solving them. We establish the algorithm's convergence properties and delve into practical aspects, including transition matrix design, the impact of bootstrap, hyperparameter sensitivity, and its utility in image classification tasks.

**Future work and limitations.** Our work does not conclusively explain the generalization benefits of using TD in deep learning settings. Furthermore, our current focus has primarily been on standard learning scenarios, with only a limited set of experiments in the imbalance scenario. It would be more intriguing to explore the utility of the probability transition matrix in a broader context, such as continual learning settings characterized by distribution shifts over time. Lastly, extending our approach to test its effectiveness with more modern NNs, such as transformers, could be of significant interest to a broader community.

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

# A  PROOFS

## A.1  Convergence under Expected Update

The convergence proofs resemble those in Bhandari et al. (2018), adapted to handle our specific case with a transformation function $f$.

As mentioned in the main text, the strategy is to bound the second and third terms of the RHS of Eq. (11). Denote $y^* = f(z^*) = f(\boldsymbol{\phi}^\top \mathbf{w}^*)$ and $y_{td}^* = f(z_{td}^*) = f(r + \boldsymbol{\phi}'^\top \mathbf{w}^*)$. The next two lemmas bound the second and third terms respectively.

**Lemma 2.** *For* $\mathbf{w} \in \mathcal{W}$, $(\mathbf{w}^* - \mathbf{w})^\top \overline{g}(\mathbf{w}) \geq \left(\frac{1}{L} - \gamma L\right) \cdot \mathbb{E}\left[(z^* - z)^2\right]$

*Proof.* Note that

$$(\mathbf{w}^* - \mathbf{w})^\top \overline{g}(\mathbf{w}) = (\mathbf{w}^* - \mathbf{w})^\top [\overline{g}(\mathbf{w}) - \overline{g}(\mathbf{w}^*)] \tag{19}$$

$$= (\mathbf{w}^* - \mathbf{w})^\top \mathbb{E}[\,[(y_{td} - y) - (y_{td}^* - y^*)]\boldsymbol{\phi}\,] \tag{20}$$

$$= \mathbb{E}[\,(z^* - z) \cdot [(y^* - y) - (y_{td}^* - y_{td})]\,] \tag{21}$$

By Lemma 1 and using the assumption that $f$ is strictly increasing, we have $(z^* - z)(y^* - y) \geq \frac{1}{L}(z^* - z)^2$. Moreover, the function $\mathbf{z} \mapsto f(\mathbf{r} + \gamma \mathbf{P}\mathbf{z})$ is $(\gamma L)$-Lipschitz so

$$\mathbb{E}\left[\,(z^* - z)(y_{td}^* - y_{td})\,\right] \leq \gamma L \cdot \mathbb{E}[(z^* - z)^2]. \tag{22}$$

Plug these two to Eq. (21) completes the proof. $\qquad \square$

**Lemma 3.** *For* $\mathbf{w} \in \mathcal{W}$, $\|\overline{g}(\mathbf{w})\|_2 \leq 2L\sqrt{\mathbb{E}\left[(z^* - z)^2\right]}$

*Proof.* To start

$$\|\overline{g}(\mathbf{w})\|_2 = \|\overline{g}(\mathbf{w}) - \overline{g}(\mathbf{w}^*)\|_2 \tag{23}$$

$$= \|\mathbb{E}[\,[(y_{td} - y) - (y_{td}^* - y^*)]\boldsymbol{\phi}\,]\|_2 \tag{24}$$

$$\leq \sqrt{\mathbb{E}[\|\boldsymbol{\phi}\|_2^2]}\sqrt{\mathbb{E}\left[[(y_{td} - y) - (y_{td}^* - y^*)]^2\right]} \qquad \text{Cauchy-Schwartz} \tag{25}$$

$$\leq \sqrt{\mathbb{E}\left[[(y^* - y) - (y_{td}^* - y_{td})]^2\right]} \qquad \text{Assumption 1} \tag{26}$$

Let $\delta \overset{\text{def}}{=} y^* - y$ and $\delta_{td} \overset{\text{def}}{=} y_{td}^* - y_{td}$, then

$$\mathbb{E}\left[(\delta - \delta_{td})^2\right] = \mathbb{E}[\delta^2] + \mathbb{E}[\delta_{td}^2] - 2\mathbb{E}[\delta\delta_{td}] \tag{27}$$

$$\leq \mathbb{E}[\delta^2] + \mathbb{E}[\delta_{td}^2] + 2|\mathbb{E}[\delta\delta_{td}]| \tag{28}$$

$$\leq \mathbb{E}[\delta^2] + \mathbb{E}[\delta_{td}^2] + 2\sqrt{\mathbb{E}[\delta^2]\mathbb{E}[\delta_{td}^2]} \qquad \text{Cauchy-Schwartz} \tag{29}$$

$$= \left(\sqrt{\mathbb{E}[\delta^2]} + \sqrt{\mathbb{E}[\delta_{td}^2]}\right)^2 \tag{30}$$

As a result,

$$\|\overline{g}(\mathbf{w})\|_2 \leq \sqrt{\mathbb{E}\left[(y^* - y)^2\right]} + \sqrt{\mathbb{E}\left[(y_{td}^* - y_{td})^2\right]} \tag{31}$$

$$\leq L\left\{\sqrt{\mathbb{E}\left[(z^* - z)^2\right]} + \sqrt{\mathbb{E}\left[(z_{td}^* - z_{td})^2\right]}\right\} \tag{32}$$

Finally, note that

$$\mathbb{E}\left[(z_{td}^* - z_{td})^2\right] = \mathbb{E}\left[\,[(r + \gamma\boldsymbol{\phi}'^\top\mathbf{w}^*) - (r + \gamma\boldsymbol{\phi}'^\top\mathbf{w})]^2\,\right] \tag{33}$$

$$= \gamma^2 \mathbb{E}\left[\,(\boldsymbol{\phi}'^\top\mathbf{w}^* - \boldsymbol{\phi}'^\top\mathbf{w})^2\,\right] \tag{34}$$

$$= \gamma^2 \mathbb{E}\left[\,(z^* - z)^2\,\right] \tag{35}$$

where the last line is because both $s, s'$ are assumed to be from the stationary distribution. Plugging this to Eq. (32) and use the fact that $\gamma \leq 1$ complete the proof. $\qquad \square$

Now we are ready to prove the main theorem:

**Theorem 1.** *[Convergence with Expected Update] Under Assumption 1-3, consider the sequence $(\mathbf{w}_0, \mathbf{w}_1, \cdots)$ satisfying Eq. (10). Let $\overline{\mathbf{w}}_T \stackrel{def}{=} \frac{1}{T}\sum_{t=0}^{T-1}\mathbf{w}_t$, $\overline{z}_T = \phi^\top\overline{\mathbf{w}}_T$ and $z^* = \phi^\top\mathbf{w}^*$. By choosing $\alpha = \frac{1-\gamma L^2}{4L^3} > 0$, we have*

$$\mathbb{E}\left[(z^* - \overline{z}_T)^2\right] \leq \left(\frac{2L^2}{1-\gamma L^2}\right)^2 \frac{\|\mathbf{w}^* - \mathbf{w}_0\|_2^2}{T} \tag{12}$$

$$\|\mathbf{w}^* - \mathbf{w}_T\|_2^2 \leq \exp\left(-T\omega\left(\frac{1-\gamma L^2}{2L^2}\right)^2\right)\|\mathbf{w}^* - \mathbf{w}_0\|_2^2 \tag{13}$$

*Proof.* With probability 1, for any $t \in \mathbb{N}_0$

$$\|\mathbf{w}^* - \mathbf{w}_{t+1}\|_2^2 = \|\mathbf{w}^* - \mathbf{w}_t\|_2^2 - 2\alpha(\mathbf{w}^* - \mathbf{w}_t)^\top\overline{g}(\mathbf{w}_t) + \alpha^2\|\overline{g}(\mathbf{w}_t)\|_2^2 \tag{36}$$

$$\leq \|\mathbf{w}^* - \mathbf{w}_t\|_2^2 - \left(2\alpha\left(\frac{1}{L} - \gamma L\right) - 4L^2\alpha^2\right)\mathbb{E}\left[(z^* - z_t)^2\right] \tag{37}$$

where $z_t \stackrel{def}{=} \phi^\top\mathbf{w}_t$. Using $\alpha = \frac{1-\gamma L^2}{4L^3} > 0$

$$\|\mathbf{w}^* - \mathbf{w}_{t+1}\|_2^2 \leq \|\mathbf{w}^* - \mathbf{w}_t\|_2^2 - \left(\frac{1-\gamma L^2}{2L^2}\right)^2\mathbb{E}\left[(z^* - z_t)^2\right] \tag{38}$$

Telescoping sum gives

$$\left(\frac{1-\gamma L^2}{2L^2}\right)^2 \times \sum_{t=0}^{T-1}\mathbb{E}\left[(z^* - z_t)^2\right] \leq \sum_{t=0}^{T-1}(\|\mathbf{w}^* - \mathbf{w}_t\|_2^2 - \|\mathbf{w}^* - \mathbf{w}_{t+1}\|_2^2) \leq \|\mathbf{w}^* - \mathbf{w}_0\|_2^2 \tag{39}$$

By Jensen's inequality

$$\mathbb{E}\left[(z^* - \overline{z}_T)^2\right] \leq \frac{1}{T}\sum_{t=0}^{T-1}\mathbb{E}\left[(z^* - z_t)^2\right] \leq \left(\frac{2L^2}{1-\gamma L^2}\right)^2\frac{\|\mathbf{w}^* - \mathbf{w}_0\|_2^2}{T} \tag{40}$$

Finally since we assume that $\|\phi(s)\|_2^2 \leq 1, \forall s$, we have $\mathbb{E}\left[(z^* - z_t)^2\right] \geq \omega\|\mathbf{w}^* - \mathbf{w}_t\|_2^2$ where $\omega$ is the maximum eigenvalue of the steady-state feature covariance matrix $\mathbf{\Sigma} = \mathbf{X}^\top\mathbf{D}\mathbf{X} = \sum_s D(s)\phi(s)\phi(s)^\top$. Therefore, Eq. (38) leads to

$$\|\mathbf{w}^* - \mathbf{w}_{t+1}\|_2^2 \leq \left(1 - \omega\left(\frac{1-\gamma L^2}{2L^2}\right)^2\right)\|\mathbf{w}^* - \mathbf{w}_t\|_2^2 \tag{41}$$

$$\leq \exp\left(-\omega\left(\frac{1-\gamma L^2}{2L^2}\right)^2\right)\|\mathbf{w}^* - \mathbf{w}_t\|_2^2 \qquad \forall x \in \mathbb{R}, 1 - x \leq e^{-x} \tag{42}$$

Repeatedly applying this bound gives Eq. (13). $\qquad\square$

### A.2  CONVERGENCE UNDER SAMPLE-BASED UPDATE

To account for the randomness, let $\sigma^2 \stackrel{def}{=} \mathbb{E}[\|g_t(\mathbf{w}^*)\|_2^2]$, the variance of the TD update at the stationary point $\mathbf{w}^*$ under the stationary distribution. Similar to Lemma 3, the following lemma bounds the expected norm of the update:

**Lemma 4.** *For $\mathbf{w} \in \mathcal{W}$, $\mathbb{E}[\|g_t(\mathbf{w})\|_2^2] \leq 2\sigma^2 + 8L^2\mathbb{E}[(z_t^* - z_t)^2]$ where $\sigma^2 = \mathbb{E}[\|g_t(\mathbf{w}^*)\|_2^2]$.*

*Proof.* To start

$$\mathbb{E}[\|g_t(\mathbf{w})\|_2^2] = \mathbb{E}[\|g_t(\mathbf{w}) - g_t(\mathbf{w}^*) + g_t(\mathbf{w}^*)\|_2^2] \tag{43}$$

$$\leq \mathbb{E}[(\|g_t(\mathbf{w}^*)\|_2 + \|g_t(\mathbf{w}) - g_t(\mathbf{w}^*)\|_2)^2] \qquad \text{Triangle inequality} \tag{44}$$

$$\leq 2\mathbb{E}[(\|g_t(\mathbf{w}^*)\|_2^2] + 2\mathbb{E}[\|g_t(\mathbf{w}) - g_t(\mathbf{w}^*)\|_2^2] \qquad (a+b)^2 \leq 2a^2 + 2b^2 \tag{45}$$

$$\leq 2\sigma^2 + 2\mathbb{E}\left[\,\|[(y_{t,td} - y_t) - (y_{t,td}^* - y_t^*)]\phi_t\|_2^2\,\right] \tag{46}$$

$$\leq 2\sigma^2 + 2\mathbb{E}\left[\,((y_{t,td} - y_t) - (y_{t,td}^* - y_t^*))^2\,\right] \qquad \text{Assumption 1} \tag{47}$$

$$= 2\sigma^2 + 2\mathbb{E}\left[\,((y_t^* - y_t) - (y_{t,td}^* - y_{t,td}))^2\,\right] \tag{48}$$

$$\leq 2\sigma^2 + 4\left(\mathbb{E}[(y_t^* - y_t)^2] + \mathbb{E}[(y_{t,td}^* - y_{t,td})^2]\right) \qquad (a-b)^2 \leq 2a^2 + 2b^2 \tag{49}$$

where $y_t^* \stackrel{\text{def}}{=} f(z_t^*) = f(\phi_t^\top \mathbf{w}^*)$ and $y_{t,td}^* \stackrel{\text{def}}{=} f(z_{t,td}^*) = f(r_t + \phi_{t+1}^\top \mathbf{w}^*)$. Note that by Lemma 1

$$\mathbb{E}[(y_t^* - y_t)^2] + \mathbb{E}[(y_{t,td}^* - y_{t,td})^2] \leq L^2\left\{\mathbb{E}\left[(z_t^* - z_t)^2\right] + \mathbb{E}\left[(z_{t,td}^* - z_{t,td})^2\right]\right\}. \tag{50}$$

Finally,

$$\mathbb{E}\left[(z_{t,td}^* - z_{t,td})^2\right] = \mathbb{E}\left[\,[(r_t + \gamma\phi_{t+1}^\top\mathbf{w}^*) - (r_t + \gamma\phi_{t+1}^\top\mathbf{w}_t)]^2\,\right] \tag{51}$$

$$= \gamma^2\mathbb{E}\left[\,(\phi_{t+1}^\top\mathbf{w}^* - \phi_{t+1}^\top\mathbf{w}_t)^2\,\right] \tag{52}$$

$$= \gamma^2\mathbb{E}\left[\,(z_t^* - z_t)^2\,\right] \tag{53}$$

where the last line is because both $s_t, s_{t+1}$ are from the stationary distribution. Combining these with Eq. (49) gives

$$\mathbb{E}[\|g_t(\mathbf{w})\|_2^2] \leq 2\sigma^2 + 4L^2(1+\gamma^2)\mathbb{E}\left[(z_t^* - z_t)^2\right] \tag{54}$$

$$\leq 2\sigma^2 + 8L^2\mathbb{E}\left[(z_t^* - z_t)^2\right]. \qquad 0 \leq \gamma \leq 1 \tag{55}$$

$$\square$$

Now we are ready to present the convergence when using i.i.d. sample for the update:

**Theorem 2.** *[Convergence with Sampled-based Update] Under Assumption 1-3, with sample-based update Eq.* (14) *and $\sigma^2 = \mathbb{E}[\|g_t(\mathbf{w}^*)\|_2^2]$. Let $\overline{\mathbf{w}}_T \stackrel{\text{def}}{=} \frac{1}{T}\sum_{t=0}^{T-1}\mathbf{w}_t$, $\overline{z}_T = \phi^\top\overline{\mathbf{w}}_T$ and $z^* = \phi^\top\mathbf{w}^*$. For $T \geq \frac{64L^6}{(1-\gamma L^2)^2}$ and a constant step size $\alpha_t = 1/\sqrt{T}, \forall t$, we have*

$$\mathbb{E}\left[(z^* - \overline{z}_T)^2\right] \leq \frac{L\left(\|\mathbf{w}^* - \mathbf{w}_0\|_2^2 + 2\sigma^2\right)}{\sqrt{T}(1 - \gamma L^2)}. \tag{15}$$

*Proof.* Note that Lemma 2 holds for any $\mathbf{w} \in \mathcal{W}$ and the expectation in $\overline{g}(\mathbf{w}) = \mathbb{E}[g_t(\mathbf{w})]$ is based on the sample $(s_t, r_t, s_{t+1})$, *regardless of the choice of* $\mathbf{w}$. Thus, one can choose $\mathbf{w} = \mathbf{w}_t$ and then $\mathbb{E}[g_t(\mathbf{w}_t)|\mathbf{w}_t] = \overline{g}(\mathbf{w}_t)$. As a result, both Lemma 2 and Lemma 4 can be applied to $\mathbb{E}[(\mathbf{w}^* - \mathbf{w}_t)^\top g_t(\mathbf{w}_t)|\mathbf{w}_t]$ and $\mathbb{E}[\|g_t(\mathbf{w}_t)\|_2^2|\mathbf{w}_t]$, respectively, in the following. For any $t \in \mathbb{N}_0$

$$\mathbb{E}[\|\mathbf{w}^* - \mathbf{w}_{t+1}\|_2^2] = \mathbb{E}[\|\mathbf{w}^* - \mathbf{w}_t\|_2^2] - 2\alpha_t\mathbb{E}[(\mathbf{w}^* - \mathbf{w}_t)^\top g_t(\mathbf{w}_t)] + \alpha_t^2\mathbb{E}[\|g_t(\mathbf{w}_t)\|_2^2] \tag{56}$$

$$= \mathbb{E}[\|\mathbf{w}^* - \mathbf{w}_t\|_2^2] - 2\alpha_t\mathbb{E}[\mathbb{E}[(\mathbf{w}^* - \mathbf{w}_t)^\top g_t(\mathbf{w}_t)|\mathbf{w}_t]] + \alpha_t^2\mathbb{E}[\mathbb{E}[\|g_t(\mathbf{w}_t)\|_2^2|\mathbf{w}_t]] \tag{57}$$

$$\leq \mathbb{E}[\|\mathbf{w}^* - \mathbf{w}_t\|_2^2] - \left(2\alpha_t\left(\frac{1}{L} - \gamma L\right) - 8L^2\alpha_t^2\right)\mathbb{E}\left[(z^* - z_t)^2\right] + 2\alpha_t^2\sigma^2 \tag{58}$$

$$\leq \mathbb{E}[\|\mathbf{w}^* - \mathbf{w}_t\|_2^2] - \alpha_t\left(\frac{1}{L} - \gamma L\right)\mathbb{E}\left[(z^* - z_t)^2\right] + 2\alpha_t^2\sigma^2 \tag{59}$$

where the last inequality is due to $\alpha_t = \frac{1}{\sqrt{T}} \leq \frac{1-\gamma L^2}{8L^3}$. Then telescoping sum gives

$$\frac{1}{\sqrt{T}}\left(\frac{1}{L} - \gamma L\right)\sum_{t=0}^{T-1}\mathbb{E}\left[(z^* - z_t)^2\right] \leq \|\mathbf{w}^* - \mathbf{w}_0\|_2^2 + 2\sigma^2 \tag{60}$$

$$\Longleftrightarrow \sum_{t=0}^{T-1}\mathbb{E}\left[(z^* - z_t)^2\right] \leq \frac{\sqrt{T}L}{1 - \gamma L^2}\left(\|\mathbf{w}^* - \mathbf{w}_0\|_2^2 + 2\sigma^2\right). \tag{61}$$

Finally, Jensen's inequality completes the proof

$$\mathbb{E}\left[(z^* - \bar{z}_T)^2\right] \leq \frac{1}{T}\sum_{t=0}^{T-1}\mathbb{E}\left[(z^* - z_t)^2\right] \leq \frac{L\left(\|\mathbf{w}^* - \mathbf{w}_0\|_2^2 + 2\sigma^2\right)}{\sqrt{T}(1 - \gamma L^2)}. \tag{62}$$

$\square$

## B  EXPERIMENT DETAILS

This section describes details for reproducing all experiments in this paper, with additional results that are not shown in the main body due to space limit.

### B.1  VERIFYING MIN-NORM EQUIVALENCE

This subsection provides details of the empirical verification in Section 4.2.

Each element of the input matrix $\mathbf{X}$ is drawn from the standard normal distribution $\mathcal{N}(0,1)$. This (almost surely) gurantees that $\mathbf{X}$ has linearly independent rows in the overparametrization regime (i.e., when $n < d$). The true model $\mathbf{w}^*$ is set to be a vector of all ones. Each label $y_i$ is generated by $y_i = \mathbf{x}_i^\top \mathbf{w}^* + \epsilon_i$ with noise $\epsilon_i \sim \mathcal{N}(0, 0.1^2)$.

We test various transition matrices $\mathbf{P}$ as shown in Table 2. For `Random`, each element of $\mathbf{P}$ is drawn from the uniform distribution $U(0,1)$ and then normalized so that each row sums to one. The `Deficient` variant is exact the same as `Random`, except that the last column is set to all zeros before row normalization. This ensures that the last state is never visited from any state, thus not having full support in its stationary distribution. `Uniform` simply means every element of $\mathbf{P}$ is set to $1/n$ where $n = 100$ is the number of training points. `Distance (Close)` assigns higher transition probability to points closer to the current point, where the element in the $i$th row and the $j$th column is first set to $\exp(-(y_i - y_j)^2/2)$ then the whole matrix is row-normalized. Finally, `Distance (Far)` uses $1 - \exp(-(y_i - y_j)^2/2)$ before normalization. The last two variants are used to see if similarity between points can play a role in the transition when using our TD algorithm.

As shown in Table 2, the min-norm solution $\mathbf{w}_{TD}$ is very close to the min-norm solution of OLS as long as $n < d$ and $\mathbf{D}$ has full support (non-deficient $\mathbf{P}$). The choice of $\mathbf{P}$ only has little effect in such cases. This synthetic experiment verifies our analysis in the main text.

### B.2  DETAILS FOR EXPERIMENT SECTION

Deep learning experiments are based on tensorflow (Abadi et al., 2015), version 2.11.0. Datasets and Code for running our experiments will be published.

**Dataset preparation.** For the CTPOS data, we have selected a subset of 10k data points for experiments to reduce training time. Regarding the Bikeshare dataset, we have performed one-hot encoding for all categorical variables and removed intuitively less relevant features such as date and year. This preprocessing results in 114 features.

**Hyperparameter settings.** For regression and binary classification tasks, we employ neural networks with two hidden layers of size 256x256 and ReLU activation functions. These networks are trained with a mini-batch size of 128 using the Adam optimizer (Kingma & Ba, 2015). In our TD algorithm, we perform hyperparameter sweeps for $\gamma \in \{0.1, 0.9\}$, target network moving rate $\tau \in \{0.01, 0.1\}$. For all algorithms we sweep learning rate $\alpha \in \{0.0003, 0.001, 0.003, 0.01\}$, except for cases where divergence occurs, such as in Poisson regression on the Bikeshare dataset, where we additionally sweep $\{0.00003, 0.0001\}$. Training iterations are set to 15k for the house data, 25k for Bikeshare, and 30k for other datasets. We perform random splits of each dataset into 60% for training and 40% for testing for each random seed or independent run. Hyperparameters are optimized over the final 25% of evaluations to avoid divergent settings. The reported results in the tables represent the average of the final two evaluations after smoothing window.

**Additional results.** Results on execution time is in table B.2, again, with increasing variance ($\sigma^2 \in \{0, 2.5, 10\}$) zero-mean Gaussian noise added to the training target from the first row to the last row.

| execution time | | |
|---|---|---|
| **TD-Reg** | **Reg-WP** | **Reg** |
| $23.7634 \pm 0.3825$ | $23.7943 \pm 0.3655$ | $23.8700 \pm 0.3603$ |
| $23.6543 \pm 0.3479$ | $23.7634 \pm 0.3448$ | $23.8907 \pm 0.3580$ |
| $23.7981 \pm 0.3661$ | $23.9577 \pm 0.3713$ | $24.0420 \pm 0.3562$ |

Table 6: Testing RMSE on execution time dataset.

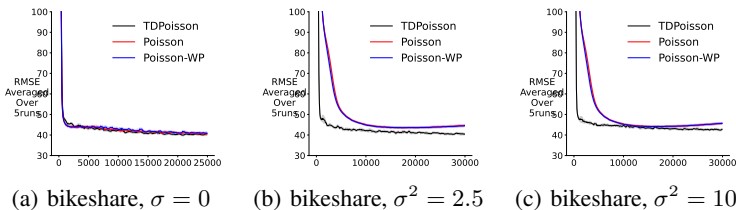

(a) bikeshare, $\sigma = 0$     (b) bikeshare, $\sigma^2 = 2.5$     (c) bikeshare, $\sigma^2 = 10$

Figure 3: Learning curves on bikeshare dataset. From left to right, the noise variance is increasing.

To gain a better understanding of how each algorithm operates, we provide additional learning curves in Figures 3 and 4. There is no statistically significant difference among the algorithms on the execution time dataset, as indicated in Table B.2, so we have omitted it. We observed that the baselines tend to overfit rapidly when noise levels are high in the house dataset. As a sanity check, we have included a fourth figure where the baselines use a learning rate that is 10 times smaller than the chosen one. However, it's worth noting that even with the reduced learning rate, overfitting still occurs, albeit at a slower pace.

**Linear function approximation result.** We did not discuss the results on the regression datasets under linear function approximation in the experiment section. This is because TD is nearly identical to OLS, even when the assumption mentioned earlier does not hold. Table 7 presents the $l_2$ norm distance between our solution and a standard OLS solution on three real-world datasets. As we explained in Section B.1, our linear TD method can be viewed as solving the left-sided sketched system $\mathbf{X}^\top \mathbf{S} \mathbf{X} \mathbf{w} = \mathbf{X}^\top \mathbf{S} \mathbf{y}$. One conjecture is that such left-sided sketching essentially provides a high-quality approximation to the original system. A separate study also observed that a left-sided random projection could yield a solution that is highly similar to the original system Pan et al. (2017).

**Binary classification.** On binary classification dataset, weather and insurance, the imbalance ratios (proportion of zeros) are around $72.45\%, 88.86\%$ respectively. We set number of iterations to 20k for both and it is sufficient to see convergence, as shown in Figure 5.

Additionally, in our TD algorithm, to prevent issues with inverse link functions, we add or subtract $1 \times 10^{-15}$ when applying them to values of $0$ or $1$ in classification tasks.

On those class-imbalanced datasets, when computing the reweighted testing error, we use the function from `https://scikit-learn.org/stable/modules/generated/sklearn.metrics.balanced_accuracy_score.html`.

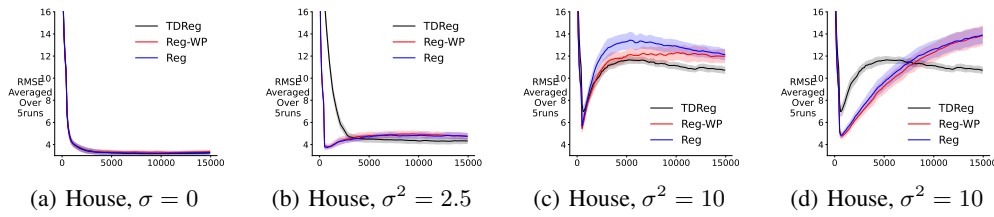

(a) House, $\sigma = 0$    (b) House, $\sigma^2 = 2.5$    (c) House, $\sigma^2 = 10$    (d) House, $\sigma^2 = 10$

Figure 4: Learning curves on house dataset. From left to right, the noise variance is increasing. The rightmost figure has the same noise as (c), with the baseline using a smaller learning rate.

| Discount rate $\gamma$ Dataset | 0.1 | 0.9 |
|---|---|---|
| House | 4.6527e-12 | 5.6187e-12 |
| CTPOS | 7.8949e-10 | 2.7222e-10 |
| Execution time | 8.9901e-10 | 2.7222e-10 |

Table 7: Distance between closed-form min-norm solutions of TD(0) and LS $\|\mathbf{w}_{TD} - \mathbf{w}_{LS}\|_2$ on real world datasets. We subsample at most 2k points to speed up computation.

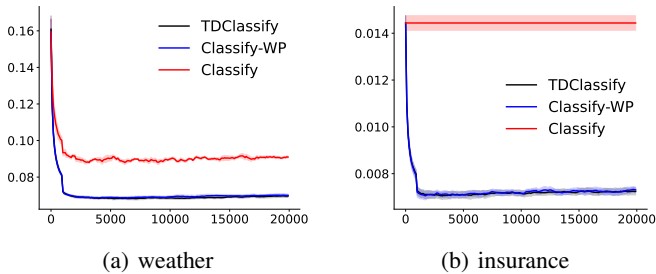

(a) weather    (b) insurance

Figure 5: Learning curves of binary classification with imbalance: balanced testing error v.s. training steps. The results have been smoothed using a 5-point window before averaging over 5 runs.

**Image datasets.** On all image datasets, Adam optimizer is used and the learning rate sweeps over $\{0.003, 0.001, 0.0003\}$, $\gamma \in \{0.01, 0.1, 0.2\}$, $\tau \in \{0.01, 0.1\}$. The neural network is trained with mini-batch size 128.

### B.3  ON THE IMPLEMENTATION OF $\mathbf{P}$

As we mentioned in Section 5, to investigate the effect of transition matrix, we implemented three types of transition matrices: $\mathbf{P}(\mathbf{x}'|\mathbf{x})$ is larger when the two points $x, x'$ 1) are similar; 2) when they are far apart; 3) $\mathbf{P}(\mathbf{x}'|\mathbf{x}) = 1/n, \forall \mathbf{x}, \mathbf{x}' \in \mathcal{X}$. We now describe the implementation details.

For first choice, given two points $(\mathbf{x}_1, y_1), (\mathbf{x}_2, y_2)$, the formulae to calculate the similarity is:

$$k(y_1, y_2) = \exp(-(y_1 - y_2)^2/v) + 0.1 \tag{63}$$

where $v$ is the variance of all training targets divided by training set size. The second choice is simply $1 - \exp(-(y_1 - y_2)^2/v)$.

Note that the constructed matrix may not be a valid probability transition matrix. To turn it into a valid stochastic matrix, we want: $\mathbf{P}$ itself must be row-normalized (i.e., $\mathbf{P}\mathbf{1} = \mathbf{1}$). To ensure fair comparison, we want equiprobable visitations for all nodes/points, that is, the stationary distribution is assumed to be uniform: $\boldsymbol{\pi} = \frac{1}{n}\mathbf{1}$.

The following proposition shows the necessary and sufficient conditions of the uniform stationary distribution property:[1]

**Proposition 3.** $\boldsymbol{\pi} = \frac{1}{n}\mathbf{1}$ *is the stationary distribution of an ergodic* $\mathbf{P}$ *if and only if* $\mathbf{P}$ *is a doubly stochastic matrix (DSM).*

*Proof.* **If**: Note that

$$\pi_j = \sum_i \pi_i p_{ij} \quad \text{and} \quad 1 = \sum_i p_{ij} \tag{64}$$

---

[1]We found a statement on Wikipedia (https://en.wikipedia.org/wiki/Doubly_stochastic_matrix), but we are not aware of any formal work that formally supports this. If such support exists, please inform us, and we will cite it accordingly.

Subtracting these two gives

$$1 - \pi_j = \sum_i (1 - \pi_i) p_{ij} \quad \text{and} \quad \frac{1 - \pi_j}{n} = \sum_i \frac{1 - \pi_i}{n} p_{ij}.$$

This last equation indicates that $\left( \frac{1-\pi_1}{n}, \frac{1-\pi_2}{n}, \cdots, \frac{1-\pi_n}{n} \right)^\top$ is also the stationary distribution of $\mathbf{P}$. Due to the uniqueness of the stationary distribution, we must have

$$\frac{1 - \pi_i}{n} = \pi_i$$

and thus $\pi_i = \frac{1}{n}, \forall i$.

**Only if**: Since $\boldsymbol{\pi} = \frac{1}{n}\mathbf{1}$ is the stationary distribution of $\mathbf{P}$, we have

$$\frac{1}{n}\mathbf{1}^\top \mathbf{P} = \frac{1}{n}\mathbf{1}^\top$$

and thus $\mathbf{1}^\top \mathbf{P} = \mathbf{1}^\top$ showing that $\mathbf{P}$ is also column-normalized. $\qquad \square$

With linear function approximation,

$$\mathbf{A} = \mathbf{X}^\top \mathbf{D}(\mathbf{I} - \gamma\lambda\mathbf{P})^{-1}(\mathbf{I} - \gamma\mathbf{P})\mathbf{X} \tag{65}$$

$$\mathbf{b} = \mathbf{X}^\top \mathbf{D}(\mathbf{I} - \gamma\lambda\mathbf{P})^{-1}(\mathbf{I} - \gamma\mathbf{P})\mathbf{y} \tag{66}$$

$$\mathbf{A} = \mathbf{X}^\top \mathbf{D}(\mathbf{I} - \gamma\mathbf{P})\mathbf{X} \tag{67}$$

$$\mathbf{b} = \mathbf{X}^\top \mathbf{D}(\mathbf{I} - \gamma\mathbf{P})\mathbf{y} \tag{68}$$

where $\mathbf{X} \in \mathbb{R}^{n \times d}$ is the feature matrix, and $\mathbf{D}$ is uniform when $\mathbf{P}$ is a DSM.

As a result, we can apply a DSM (Wang et al., 2010) projection method to our similarity matrix.

