# OpenReview forum: "Generalized Temporal Difference Learning Models for Supervised Learning"
_ICLR.cc/2024/Conference — Submitted to ICLR 2024_

### Official Review · Reviewer_WyFN · 2023-10-29

**Soundness:** 4 excellent
**Presentation:** 3 good
**Contribution:** 3 good
**Rating:** 8
**Confidence:** 3

**Summary:**

The authors describe how to model the data sampling/generation
process, in a supervised learning framework, as a Markov reward
process. This contrasts with the established i.i.d. data assumption
used in most classifiers.
The authors consider that the data points are originated from a Markov
reward process, where the policy is fixed and the goal is to perform a
policy evaluation.
They introduce a generalized temporal-difference (TD) algorithm for this.

They proposed a mapping between the training features (X) and the
states in RL, and between the target variable (y) and the state value
function. Under this framework, the reward function can be estimated
from the differences between the output variables from successive times.
They show that under a TD framework, the updating rule can bootstrap
with the output variable of the next time.

They also provide a generalization to different types of data
by incorporating logits. They provide convergence proofs and several
empirical studies showing different aspects of the proposed approach.

**Strengths:**

- Relate two different learning paradigms
- Proposes a theoretical framework
- Present several theoretical results

**Weaknesses:**

- The paper is not easy to read

**Questions:**

The proposed approach seems only applicable to supervised
learning. How could it be extended to unsupervised learning where the
is not a clear target output.

---

> ### Author Response · Authors · 2023-11-11
>
> Thank you for your positive feedback and valuable insights. As we mentioned in our introduction, we view this work as an important and initial step towards unifying various learning settings, and unsupervised learning is a direction that we will definitely explore in the future.

---

### Official Review · Reviewer_ZxKx · 2023-10-29

**Soundness:** 2 fair
**Presentation:** 3 good
**Contribution:** 1 poor
**Rating:** 3
**Confidence:** 4

**Summary:**

The paper applies reinforcement learning to solve conventional regression/supervised learning problems. In particular, the author(s) focused on classical generalized linear models and reformulated the parameter estimation with a reinforcement learning (RL) framework. This allows to apply the classical on-policy evaluation algorithm in RL to solve supervised learning. In theory, the author(s) established the convergence properties of the estimated parameters. Empirically, they further extended their methodology to deep learning.

**Strengths:**

As far as I can see, the strengths of the paper can be summarized as follows:

**Originality**: The idea to apply RL to solve supervised learning is relatively less explored in the literature. As the author(s) commented in the paper, existing algorithms are particularly designed for specific types of problems. To my knowledge, similar ideas have not been proposed in the literature.

**Quality**: Compared to the existing RL algorithms to solve supervised learning problems, the solution the author(s) proposed is general and is applicable to a wide range of regression problems covering parameter estimation in both classical generalized linear models and deep learning.

**Clarity**: The paper is easy to follow. The writing is generally clear.

**Weaknesses:**

In my assessment, the primary limitation of this paper stems from the effectiveness of the proposed method. From my interpretation, the recommended RL algorithm appears to be less efficient than traditional supervised learning algorithms, particularly from a theoretical standpoint. This issue considerably constrains the practicality of the presented methodology, and I will provide a comprehensive explanation below.

Let's initiate the discussion with the ordinary least square (OLS) regression problem, which is introduced at the onset of Section 3. The conventional OLS method calculates the following estimator $\widehat{\omega}_{\textrm{SL}}$ for $\omega^*$, the oracle value in the linear model,

$$\widehat{\omega}_{\textrm{SL}}=(\sum_i x_i x_i^\top)^{-1} (\sum_i x_i y_i).$$

The mean squared error (MSE) of this estimator can be shown to be equivalent to

$$n\textrm{MSE}(\widehat{\omega}_{\textrm{SL}})=\sigma^2 \textrm{trace}(\Sigma^{-1}),$$

where $n$ denotes the sample size, $\Sigma= \mathbb{E} x_i x_i^\top$ and $\sigma^2$ denote the error variance $\mathbb{E} (y_i-x_i^\top \omega^*)^2$. It is widely recognized that the OLS estimator is the Best Linear Unbiased Estimator (BLUE), minimizing the MSE among all unbiased estimators.

Moving on to the RL estimator, as depicted in Equation (4), the resulting estimator $\widehat{\omega}_{\textrm{RL}}$ can be expressed as

$$\Big[\sum_i x_i (x_i-\gamma x_{i+1})^\top\Big]^{-1} \Big[\sum_i x_i (y_i - \gamma y_{i+1})\Big].$$

Assuming both $x_i$ and $y_i$ are of mean zero, its MSE can be shown to equal to

$$n\textrm{MSE}(\widehat{\omega}_{\textrm{RL}})=(1+\gamma^2)\sigma^2 \textrm{trace}(\Sigma),$$

which is strictly larger than the SL-based estimator as long as $\gamma>0$. Additionally, the larger the discounted factor $\gamma$, the larger the MSE. This observation demonstrates that when tailored to linear models, the RL-based estimator exhibits statistical inefficiency in comparison to the SL-based estimator, particularly from a theoretical standpoint. A thorough examination of the closed-form expression for the RL estimator​ sheds light on the root cause of this inefficiency. In the RL estimator, the target variable encompasses both the current outcome $y_i$ (which embeds the true signal) and the future outcome $y_{i+1}$. It is crucial to note that the latter is independent of
$x_i$ and can be perceived as an additional source of error. By adopting the RL framework, we inadvertently introduce extra noise to the outcome, ultimately compromising the efficiency of the resulting estimator.

Further considering the more extensive Generalized Linear Model (GLM), it is established that the SL-based estimator, when computed via Newton-type algorithms such as Fisher scoring, achieves efficiency, attaining the Cramer-Rao (CR) lower bound. I suspect that similar disparities are present in Generalized Linear Models (GMLs), where the RL-based estimator may not reach the CR lower bound, exhibiting reduced efficiency, due to the additional noise introduced in the target variable.

It is crucial to note that the above analysis predominantly pertains to estimators derived through Newton-type algorithms. When gradient-type algorithms are employed in conjunction with the Polyak-Ruppert averaging scheme, the resulting estimators exhibit the same asymptotic distributions as those computed via Newton's method. This leads us to the same conclusion.

**Questions:**

Please refer to the weakness section.

---

> ### Author Response · Authors · 2023-11-11
>
> Thank you for reading our paper, providing your insights and recognizing the originality, quality and clarity of our work.
>
> We also value the fact that you initiated the discussion by writing down the theoretical insight and explicitly highlighted your concerns regarding the practical utility of our approach.
>
> The difference between your theoretical insight and ours in Section 4.2 likely stems from that Assumption 4 does not hold in your example. As demonstrated in Section 4.2, the two solutions of SL and RL are precisely the same under Assumption 4 given a dataset. Consequently, the Mean Squared Error (MSE) of the two approaches should be exactly equal. We would like to respectfully argue that our assumption is closer to practical scenarios than your example. Assumption 4 can generally be considered true in the overparameterized regime, where deep learning resides (refer to the neural tangent kernel and many of its follow-ups), especially as the model size increases. Please let us know if you find any issue with our calculation about equivalence of TD and SL solutions.
>
> Understanding the theoretical properties of our solution beyond the linear case (e.g., with respect to the CRLB) is certainly an important research direction. However, it requires a nontrivial effort and is worth a separate work in itself. We will continue investigating this in the future.
>
> Perhaps it would be helpful to further clarify how we discuss the practical utility of our TD in the paper. Section 4.2 states that under Assumption 4, our method identifies the same solution as conventional regression. We then provide empirical results on both synthetic (Table 2) and real-world datasets (Table 7), demonstrating that the two solutions can be extremely close even when the assumption is not satisfied. Furthermore, for showing practical utility, we spend the whole experiments section in deep learning, we discovered that our approach exhibits competitive generalization performance and even outperforms traditional methods in special cases.
>
> We hope these empirical results sufficiently demonstrate the potential utility of our approach, and you would agree with us that the absence of the justification you mentioned should not be a reason to overlook or invalidate empirical results. There is no fundamental contradiction between them.
>
> We would be happy to continue the discussion with you if you have further concerns or insights to share.

---

> > ### Comment · Reviewer_ZxKx · 2023-11-13
> >
> > Many thanks for the clarification. I totally agree that the proposed estimator is designed to approximate the performance of the SL-based estimator under the stated assumption. However, I remain skeptical about the utility of the method. It prompts the question: What is the advantage of introducing a new methodology that only parallels the performance of an established method under certain assumptions and may underperform when those assumptions do not hold?

---

> > > ### Author Response · Authors · 2023-11-13
> > >
> > > Thank you for your response. We politely disagree with the characterization of our method as merely parallel or underperforming. Our experiments confirm that in the more common over-parameterized regime of deep learning TD demonstrates advantages over OLS.
> > >
> > > Requiring an algorithm to be better across all settings is likely unrealistic.
> > >
> > > We believe our work takes an early step in unifying two learning paradigms, potentially bridging recent advancements in modern RL with traditional SL. Considering TD as a fundamental algorithm, our approach opens avenues for exploring additional RL methods like GTD, TDC, and potentially redefining approaches in transfer learning or nonstationary supervised learning settings. Suppressing an initial exploration that is reasonably justified also extinguishes all possibilities afterward.
> > >
> > > Regarding your MSE derivation, we are unclear how you get that MSE of TD’s solution. Note that there is a transition matrix in the inverse when you write into matrix form. We would appreciate further derivation details of the RL solution’s MSE to facilitate a comprehensive discussion.

---

> > > > ### Comment · Reviewer_ZxKx · 2023-11-13
> > > >
> > > > Thank you for your follow-up. I would like to clarify that my expectation is not for your algorithm to outperform others in every possible scenario. Nonetheless, I am concerned that I have not identified any scenarios where your algorithm demonstrates superior performance. As indicated in my earlier comments, the introduction of additional noise into the outcomes tends to render the resultant estimator less efficient. While you have shown that your method can match the efficiency of classical methods under certain conditions, it remains unclear in which scenarios it would outperform them. Although you have demonstrated the empirical benefits of your method in specific instances, it is generally possible to find cases where one method outperforms another. I will provide you with the detailed calculations later.

---

> > > > > ### Comment · Reviewer_ZxKx · 2023-11-15
> > > > >
> > > > > As mentioned earlier, the solution to the TD estimator $\widehat{\omega}_{\textrm{RL}}$ can be expressed as
> > > > >
> > > > > $$\Big[\sum_i x_i (x_i-\gamma x_{i+1})^\top\Big]^{-1} \Big[\sum_i x_i (y_i - \gamma y_{i+1})\Big].$$
> > > > >
> > > > > Consequently, the residual $\widehat{\omega}_{\textrm{RL}}-\omega^*$ is given by
> > > > >
> > > > > $$\Big[\sum_i x_i (x_i-\gamma x_{i+1})^\top\Big]^{-1} \Big[\sum_i x_i (y_i -x_i^\top\omega^*- \gamma y_{i+1}+\gamma x_{i+1}^\top\omega^*)\Big]=\Big[\frac{1}{n}\sum_i x_i (x_i-\gamma x_{i+1})^\top\Big]^{-1} \frac{1}{n}\Big[\sum_i x_i (e_i- \gamma e_{i+1})\Big],$$
> > > > >
> > > > > where $e_i$ denote the error $y_i-x_i^\top \omega^*$. The first term on the right-hand-side is convergent to $\mathbb{E} x_i x_i^\top - \gamma \mu_x \mu_x^\top=\Sigma -\gamma \mu_x \mu_x^\top$ where $\mu_x=\mathbb{E} x_i$ whereas the second term is $O_p(n^{-1/2})$. Consequently, $\widehat{\omega}_{\textrm{RL}}-\omega^*$ is root-n consistent to
> > > > >
> > > > > $$\Big[\Sigma-\gamma\mu_x \mu_x^\top\Big]^{-1} \frac{1}{n}\Big[\sum_i x_i (e_i- \gamma e_{i+1})\Big].$$
> > > > >
> > > > > When the error residual $e_i$ is independent of $x_i$, the covariance matrix of the above expression is given by
> > > > >
> > > > > $$\frac{1}{n}\Big[\Sigma-\gamma\mu_x \mu_x^\top\Big]^{-1} \textrm{Cov}[x_i (e_i- \gamma e_{i+1})] \Big[\Sigma-\gamma\mu_x \mu_x^\top\Big]^{-1}=\frac{(1+\gamma^2)\sigma^2}{n} \Big[\Sigma-\gamma\mu_x \mu_x^\top\Big]^{-1} \Sigma \Big[\Sigma-\gamma\mu_x \mu_x^\top\Big]^{-1}.$$
> > > > >
> > > > > Notice that the above expression is miniminized when $\gamma=0$, which corresponds to the OLS solution. Meanwhile, it is lower bounded by
> > > > >
> > > > > $$\frac{(1+\gamma^2)\sigma^2}{n}\Sigma^{-1}.$$
> > > > >
> > > > > Consequently, the MSE of $\widehat{\omega}_{\textrm{RL}}$ is asymptotically lower bounded by $$\frac{(1+\gamma^2)\sigma^2}{n}\textrm{trace}(\Sigma^{-1}),$$ which is strictly larger than that of the OLS solution when $\gamma>0$.

---

> > > > > > ### Author Response · Authors · 2023-11-16
> > > > > >
> > > > > > Thank you for your very careful review and for writing your insights. However, the derivation is problematic.
> > > > > >
> > > > > > The critical step ''the first term on the right-hand-side is convergent to $\mathbb{E} x_i x_i^\top - \gamma \mu_x \mu_x^\top=\Sigma -\gamma \mu_x \mu_x^\top$" relies on the assumption that $x_i$ and $x_{i+1}$ are independent (or at least uncorrelated), a condition that is NOT true in RL, where the next data point $x_{i+1}$ is sampled based on the previous point $x_i$ according to the transition matrix.

---

> > > > > > > ### Comment · Reviewer_ZxKx · 2023-11-16
> > > > > > >
> > > > > > > I apologize if I've overlooked anything. When applying RL to supervised learning, it's important to note that the data in supervised learning is generated i.i.d. Using a different method won't change this fundamental aspect of the data generation process...

---

> > > > > > > > ### Author Response · Authors · 2023-11-16
> > > > > > > >
> > > > > > > > No, $x_t$ and $x_{t+1}$ are **not** i.i.d. in our algorithm. The sampling process can be described as follows: first, sample $x_t$ from $D$, where $D$ is the stationary distribution of the Markov reward process; then, sample $x_{t+1}$ from $P(\cdot | x_t)$. Thus, $x_{t+1}$ depends on $x_t$. We strongly suggest you double-check Section 3 for more details on our formulation.
> > > > > > > >
> > > > > > > > With this clarification, we hope you reevaluate the significance of our contribution. We appreciate your efforts in conducting the analysis.

---

> > > > > > > > > ### Comment · Reviewer_ZxKx · 2023-11-16
> > > > > > > > >
> > > > > > > > > Dear author(s), thank you for your response. I would like to make some clarifications. I tried to follow the notations in your manuscript. $x_i$ and $y_i$ I used denote the $i$th sample in the supervised learning data. It differs from $x^{(t)}$, the covariates you sampled at the $t$th time point in the RL algorithm. While I agree that $x^{(t+1)}$ might depend on $x^{(t)}$ given the dataset, but $x_i$ and $x_{i+1}$ are independent. Meanwhile, in your experiment, you mentioned that "In TD algorithms, unless otherwise specified, we use a fixed transition probability matrix with all entries set to 1/n. This choice simplifies memory management, generation, and sampling processes." In this case, even $x^{(t+1)}$ and $x^{(t)}$ are independent.
> > > > > > > > >
> > > > > > > > > I also would like to mention that I have spent a lot of time in providing the feedback and responding to your queries. My purpose is to let you know my understanding of your proposal. Of course, I might be wrong, but I hope you can understand the logic behind my calculation so that you can also go through the calculations yourselves to compare different algorithms.
> > > > > > > > >
> > > > > > > > > In the RL algorithm you implemented, you use a Markov process to resample all the observations. Suppose at time $t$, let $P(\bullet|x^{(t)})$ denote the transition kernel, according to your notation. This yields a conditional probability density functions of all data points given $x^{(t)}$. In my previous calculation, I assume this transition kernel is degenerate, i.e., for each $x_i$, $P(\bullet|x_i)$ will produce a deterministic sample. Sorry that I simplified the problem earlier but did not mention all the assumptions. Below, I will relax this assumption. However, the conclusion is similar: the proposed estimator is no better than the OLS estimator generally.
> > > > > > > > >
> > > > > > > > > First, by looking at table 2, I notice that the random or uniform transition performs the best. To start with, let me consider the case with a uniform transition function. In this case, we can similarly show that the error of the TD estimator $\widehat{\omega}_{\textrm{RL}}-\omega^*$ can be represented by
> > > > > > > > > $$\Big[n^{-1}\sum_i x_i (x_i-n^{-1}\sum_j \gamma x_j)^\top\Big]^{-1} \Big[n^{-1}\sum_i x_i (e_i - n^{-1}\sum_j \gamma e_j)\Big].$$
> > > > > > > > > Here, all the superscripts index the samples, and $x_i$ and $x_j$ are independent whenever $i\neq j$. The first term is again to converge to $(\Sigma-\gamma \mu_x \mu_x^\top)^{-1}$ whereas the co-variance of the second term is asymptotically equivalent to $n^{-1}(\Sigma-2\gamma \mu_x \mu_x^\top+\gamma^2 \mu_x\mu_x^\top)$. This yields the following covariance matrix for the estimated TD estimator
> > > > > > > > >
> > > > > > > > > $$\frac{1}{n}(\Sigma-\gamma\mu_x\mu_x^\top)^{-1} (\Sigma-2\gamma \mu_x \mu_x^\top+\gamma^2 \mu_x\mu_x^\top) (\Sigma-\gamma\mu_x\mu_x^\top)^{-1}.$$
> > > > > > > > >
> > > > > > > > > When the covariance of $x_i$ is not degenerate, $\mu_x\neq 0$ and $\gamma>0$, it is strictly larger than the co-variance of the OLS estimator.
> > > > > > > > >
> > > > > > > > > Second, moving to general transition functions, it would be very difficult to analytically compare the MSE. However, notice that when the transition matrix does not involve $y$, the TD estimator is still a linear function of the response. According to the Gauss Markov theorem, the MSE of the OLS estimator would be the smallest. The intuition is again, as I mentioned earlier, to implement the RL estimator, it inevitably introduce extra noise to the outcome, leading to a lower efficiency of the resulting estimator. When the transition function involves $y$, the TD estimator would be nonlinear. However, in that case, even its consistency is not guaranteed.

---

> ### Author Response · Authors · 2023-11-17
>
> Thank you for your diligence in reviewing our paper. We acknowledge the considerable time and effort you invested, and we appreciate that.
>
> Regarding your statement "but $x_i$ and $x_{i+1}$ are independent". If this is the case, then your expression $\Big[\sum_i x_i (x_i-\gamma x_{i+1})^\top\Big]^{-1} \Big[\sum_i x_i (y_i - \gamma y_{i+1})\Big]$ is **not** our RL solution since our estimator requires that $x_i$ and $x_{i+1}$ are related by the transition kernel.
>
> Regarding Table 2, it is not surprising that the uniform $P$ performs the best, as the synthetic data is generated by uniform sampling (see Appendix B.1). In fact, this example verifies that it would be best to use a $P$ that matches the data generating process (so simply assume P is uniform in theoretical analysis in the underparameterized case seems not reasonable). Upon request, we can design another synthetic dataset to show when uniform is not the best. The key message here is to show that as long as the feature dimension is reasonably large, our TD solution matches the OLS solution irrespective of $P$, as a verification of our equivalence analysis.
>
> Regarding our implementation of using a uniform transition matrix in Section 5, it aligns with our theoretical analysis of OLS and TD in the overparameterized regime (deep NN experiments), leading us to opt for a convenient implementation. A simple choice can lead to competitive performance for the benchmark datasets, showing the promising practical utility of our method.
>
> We want to further clarify that, under OLS assumptions, yes, OLS is a BLUE. However, applying the assumptions from OLS to analyze our TD solution is not meaningful and not fair.
>
> To draw an analogy, consider someone assuming Laplacian distributed data and proposing to minimize the absolute loss. Criticizing this solution's performance in terms of MSE compared to OLS is not interesting, since they rely on different assumptions.
>
> Finally, could you explain "The intuition is again, as I mentioned earlier, to implement the RL estimator, it inevitably introduces extra noise to the outcome"? Particularly, what noise are you referring to?
>
> Again, we do appreciate your efforts and your active discussion with us.

---

> > ### Author Response · Authors · 2023-11-21
> >
> > We assume that your perception of TD's solution introducing more noise is attributed to the construction of the reward, which requires two samples’ training targets. The variance calculation involving these two variances may introduce more noise. However, this reasoning again, is based on the iid samples. Note that when two samples are correlated, their summation could result in lower variance.
> >
> > Looking at it from this perspective, in response to your earlier question about when the algorithm would perform well, one could argue that it happens when the algorithmic assumption closely aligns with the ground truth of data generation distribution. However, determining the ground truth is more of a personal belief, as it is rarely known in practical scenarios.
> >
> > Please inform us if we have adequately addressed your concerns.

---

> > > ### Comment · Reviewer_ZxKx · 2023-11-21
> > >
> > > Dear Author(s), I noticed that your approach essentially applies RL techniques to solve an SL problem. At the beginning of Section 3, you reference handling the same dataset as in SL. As commonly observed in SL applications, and as discussed in Section 2.1 of your paper, these datasets are typically i.i.d. As I have mentioned earlier, it's important to note that applying a different algorithm, such as RL in this case, does not change the underlying data distribution. While $(x^{(t)}, y^{(t)}: t\ge 1)$ may be independent over $t$ given the dataset $\mathcal{D}$, the individual data points $(x_i, y_i: 1\le i\le n)$ still retain their independence.

---

> > > > ### Author Response · Authors · 2023-11-21
> > > >
> > > > Dear reviewer, thank you for your quick response.
> > > >
> > > > “you reference handling the same dataset as in SL … ” --- Yes.
> > > >
> > > > “the individual data points still retain their independence.” --- **No, independence is an algorithmic assumption specific to certain algorithms; it is not the ground truth of the data generation**. OLS relies on the iid Gaussian assumption for Maximum Likelihood Estimation derivation and Mean Squared Error (MSE) analysis.

---

### Official Review · Reviewer_oS7Z · 2023-10-30

**Soundness:** 2 fair
**Presentation:** 3 good
**Contribution:** 2 fair
**Rating:** 6
**Confidence:** 3

**Summary:**

This paper generalizes the temporal difference (TD) learning models for supervised learning with dataset treated as interconnected. To this end, this paper introduces a generalized TD learning algorithm and establishes theoretical convergence guarantees under both expected update and sample-based update. In addition, this paper also conducts experiments, which shows the generalization performance of the TD learning algorithm.

**Strengths:**

1. This paper proposes a new generalized TD learning algorithm and establishes theoretical convergence guarantees.

2. This paper conducts experiments which shows the generalization performance of the TD learning algorithm.

**Weaknesses:**

1. Assumption 2 seems to be a little strong to require $f$ is invertible. In addition, how will this assumption be when the logit $z$ is a vector? For example, when $f$ is a softmax function?

2. The algorithm 1 requires the knowledgement of transition $P$, which seems not realistic.

Minors:

In assumption 1, $D(s)$ comes without definitions and explanation.

**Questions:**

1. I wonder how is lemma 1 derived? I think lemma 1 requires that the function $f$ is $L$-Lipschitz continuous. It seems that this can not be implied by assumption 1 and 2.

2. I want to understand whether the transition kernel $P$ will influence the theoretical results. I do not find any term related to $P$ in theorems.

---

> ### Author Response · Authors · 2023-11-11
>
> Thank you for your insights and comments. We address your questions below.
>
> Please note that the Assumption 2 about invertible $f$ is not strong, as it is a common property of the inverse link functions used by generalized linear models, which can be found in Table 2.1 of McCullagh \& Nelder (1989) and the [wiki page of GLMs](https://en.wikipedia.org/wiki/Generalized_linear_model#Link_function). Note that almost all distributions corresponding to popular loss functions satisfy this assumption.
> Additionally, when the logits form a vector $\mathbf{z}$, Assumption 1 is translated into the requirement that the antiderivative $F$ (i.e., $\nabla F=f$) is a continuously-differentiable, strictly convex function. As a concrete example, when $f$ is the softmax transformation, $F$ is the log-sum-exp function.
> One can refer to Banerjee et al. (2005) for more details for the multivariate cases.
>
> For Lemma 1, one can see that when the features $\phi$ and parameters $w$ are bounded, effectively the domain and image of $f$ will also be bounded. Thus there exists a constant $L=\max(L_f, L_{f^{-1}})$ that satisfies the lemma where $L_f, L_{f^{-1}}$ are the Lipschitz constants of $f$ and $f^{-1}$ for the domain/image respectively. While we didn't provide a formal proof here due to its simplicity, we will make sure to clarify this point in revision.
>
> Regarding the transition probability, it is designed by us, and therefore, we have complete knowledge of it. In theory, the convergence result only necessitates $P$ to be ergodic, which is a weak assumption commonly used by many previous works in TD-like algorithms' convergence. In practice, Section 4.2 and 5 demonstrate the ease of designing such a matrix, and a simple choice could be a uniformly constant matrix.

---

> > ### Comment · Reviewer_oS7Z · 2023-11-15
> >
> > Many thanks for the reponses, most of my concerns are resolved. But your proof for lemma 1 is not so clear to me. The above proof wants to claim that continuous functions are Lipstchitz continuous on bounded set. It seems not correct. Take $\sqrt{x}$ over $[0,1]$ as a counter example. It is obviously not Lipstchitz continuous. But I think it may not be a great problem if the authors can solve this with slightly stronger assumptions and provide justification of the assumptions.
> >
> > I have also read through the reviews of Reviewer ZxKx, and agreed with the issue pointed out by Reviewer ZxKx: given the calculation of MSE of RL estimator in the reviews of Reviewer ZxKx, the RL estimator is always no better than the SL estimator. I will also keep an eye on the further discussion.
> >
> > If these issues are solved, I would like to raise my scores.

---

> > > ### Author Response · Authors · 2023-11-16
> > >
> > > Thank you for your careful review and response.
> > >
> > > It seems we accidentally omitted the bounded derivative assumption during our submission. This assumption holds for commonly used functions such as linear, exponential, sigmoid, etc., under the bounded domain conditions. For an easy reference, please see Proposition 3 from [this document](https://users.wpi.edu/~walker/MA500/HANDOUTS/LipschitzContinuity.pdf).
> > >
> > > Regarding your concern about the MSE issue raised by another reviewer, we have already addressed this. Please see details in that thread. As a quick response here, the derivation of the other reviewer relied on some specific assumption, which is not suitable under our setting.
> > >
> > > We hope this clarifies any concerns you may have had. If you still think the soundness is not satisfactory, please let us know where you think there is an issue.

---

### Official Review · Reviewer_9RLa · 2023-10-31

**Soundness:** 3 good
**Presentation:** 3 good
**Contribution:** 3 good
**Rating:** 8
**Confidence:** 3

**Summary:**

In the traditional statistical learning setting, data is assumed to be independently and identically distributed (i.i.d.) according to some unknown probability distribution. Supervised algorithms such as generalized linear models are constructed by making assumptions about the distribution of the response variable given the independent variables. The authors propose an assumption where the data is interconnected and the sampling process is a Markov reward process. This turns the supervised learning problem into an on-policy evaluation problem in reinforcement learning, which the authors address by utilizing a generalized temporal difference (TD)
learning algorithm. They theoretically prove the convergence of these algorithms under linear function approximation and explore the relationship between the generalized TD solution and the original linear regression solution. Through their experiments, they compare the generalized TD algorithm with other baseline models in regression, binary classification, and image classification tasks.

**Strengths:**

I appreciate that the authors provided clear background information on supervised learning and temporal difference learning before outlining the theoretical proofs. I also appreciate that the authors also provided limitations of their work within the main text.

**Weaknesses:**

I believe the authors could have provided a few more datasets for the regression problem and the binary classification class imbalance problem.

**Questions:**

N/A

---

> ### Author Response · Authors · 2023-11-11
>
> Thank you for reading our paper and the positive comments.
>
> Please note that we do have additional results on regression (Table 6) in the Appendix B.2. As requested, we added another classification experiment (cancer dataset in Table 4), but they do not alter any conclusion. Now we have 4 regression datasets, 3 binary classifications, 4 popular image classification tasks in total. For regression and binary classification, we studied a variety of settings including different noise settings, and transition probability matrix.
>
> We hope you find our experiments are well-designed, including careful study of each design choice of our algorithm, and the datasets are representative and widely recognized as popular benchmarks.

---

> ### Comment · Reviewer_9RLa · 2023-12-02
> **Response to Authors**
>
> Thank you for your response. Since you have added the new experiments, I have decided to raise my score.

---

### Meta-Review · Area_Chair_R3m6 · 2023-12-10

**Metareview:**

This paper proposes a generalized TD-learning based method for supervised learning style tasks -- the idea is to model supervised learning as a TD-learning problem where datapoints come one after each other sequentially. I think the estimator proposed is greatly inspired from linear TD-learning and the paper provides a theoretical analysis of this estimator. And they show that when the transition matrix is chosen appropriately, their approach boils down to simple supervised learning.

That said, I do not find the experiments and the analysis convincing enough. I share concerns of Reviewer ZxKx -- while I agree that this paper provides a way to go beyond the I.I.D. assumption in SL, it is still important showing concrete theoretical results showing that simply doing ERM when the datapoints are not I.I.D. drawn from a given distribution is worse than TD-learning with a suitable choice of P. In other words, the transition matrix P is a free variable, and we can choose the best P for any problem. But still it is useful to understand conditions under which theoretically that TD-learning can do better than ERM. Right now the analysis in Section 4.2 shows equivalence to SL, but does not show when it can be better strictly. The authors have a compelling intuition that it can be better, but it is not shown in the paper anywhere.

With regards to the empirical results -- I think the gap in image classification experiments is just too small to make any conclusion, and the other datasets appear non-standard. Moreover, with P = 1/n and I.I.D. data, it is unclear why TD-learning should be better in these experiments.

Overall, while I like the idea of the paper (and so do the reviewers), I think the paper is not complete enough and some important questions are still unanswered. I encourage the authors to address these questions and submit to the next conference.

**Justification For Why Not Higher Score:**

Reviewer ZxKx's concerns are important but unaddressed, it is unclear why the paper's experimental results are not just an artifact of noise in experimentation, and the paper does not convincingly demonstrate its claims.

**Justification For Why Not Lower Score:**

N/A

---

### Decision · Program_Chairs · 2024-01-16

Reject